# Gaze Point Tracking Based on a Robotic Body–Head–Eye Coordination Method

**DOI:** 10.3390/s23146299

**Published:** 2023-07-11

**Authors:** Xingyang Feng, Qingbin Wang, Hua Cong, Yu Zhang, Mianhao Qiu

**Affiliations:** 1Army Academy of Armored Forces, Beijing 100072, China; m17695720631@163.com (X.F.); 18911025632@163.com (H.C.); zhangyuzgy2018@163.com (Y.Z.); 2Research Center of Precision Sensing and Control, Institute of Automation, Chinese Academy of Sciences, Beijing 100190, China; maokang94@163.com

**Keywords:** bionic eyes, gaze point tracking, gaze point approaching, body–eye–head coordination, 3D coordinates

## Abstract

When the magnitude of a gaze is too large, human beings change the orientation of their head or body to assist their eyes in tracking targets because saccade alone is insufficient to keep a target at the center region of the retina. To make a robot gaze at targets rapidly and stably (as a human does), it is necessary to design a body–head–eye coordinated motion control strategy. A robot system equipped with eyes and a head is designed in this paper. Gaze point tracking problems are divided into two sub-problems: in situ gaze point tracking and approaching gaze point tracking. In the in situ gaze tracking state, the desired positions of the eye, head and body are calculated on the basis of minimizing resource consumption and maximizing stability. In the approaching gaze point tracking state, the robot is expected to approach the object at a zero angle. In the process of tracking, the three-dimensional (3D) coordinates of the object are obtained by the bionic eye and then converted to the head coordinate system and the mobile robot coordinate system. The desired positions of the head, eyes and body are obtained according to the object’s 3D coordinates. Then, using sophisticated motor control methods, the head, eyes and body are controlled to the desired position. This method avoids the complex process of adjusting control parameters and does not require the design of complex control algorithms. Based on this strategy, in situ gaze point tracking and approaching gaze point tracking experiments are performed by the robot. The experimental results show that body–head–eye coordination gaze point tracking based on the 3D coordinates of an object is feasible. This paper provides a new method that differs from the traditional two-dimensional image-based method for robotic body–head–eye gaze point tracking.

## 1. Introduction

When the magnitude of a gaze is too large, human beings change the orientation of their head or body to assist their eyes in tracking targets because saccade alone is insufficient to keep a target at the center region of the retina. Studies on body–head–eye coordination gaze point tracking are still rare because the body–head–eye coordination mechanism of humans is prohibitively complex. Multiple researchers have investigated the eye–head coordination mechanism, binocular coordination mechanism and bionic eye movement control. In addition, researchers have validated the eye–head coordination models on eye–head systems. This work is significant for the development of intelligent robots for human–robot interaction. However, most of these methods are based on the principle of neurology, and their further developments and applications may be limited by people’s understanding of human processes. However, binocular coordination based on the 3D coordinates of an object is simple and practical, as verified by our previous paper [1].

When the fixation point transfers greatly, the head and eyes should move in coordination to accurately shift the gaze to the target. Multiple studies have built models of eye–head coordination based on the physiological characteristics of humans. For example, Kardamakis A A et al. [2] researched eye–head movement and gaze shifting. The best balance between eye movement speed and the duration time was sought, and the optimal control method was used to minimize the loss of motion. Freedman E G et al. [3] studied the physiological mechanism of coordinated eye–head movement. However, they did not establish an engineering model. Nakashima et al. [4] proposed a method for gaze prediction that combines information on the head direction with a saliency map. In another study [5], the authors presented a robotic head for social robots to attend to scene saliency with bio-inspired saccadic behaviors. The scene saliency was determined by measuring low-level static scene information, motion, and prior object knowledge. Law et al. [6] described a biologically constrained architecture for developmental learning of eye–head gaze control on an iCub robot. They also identified stages in the development of infant gaze control and proposed a framework of artificial constraints to shape the learning of the robot in a similar manner. Other studies have investigated the mechanisms of eye–head movement for robots and achieved satisfactory performance [7,8].

Some application studies based on coordinated eye–head movement have been carried out in addition to the mechanism research. For example, Kuang et al. [9] developed a method for egocentric distance estimation based on the parallax that emerges during compensatory head–eye movements. This method was tested in a robotic platform equipped with an anthropomorphic neck and two binocular pan–tilt units. Reference [10]’s model is capable of reaching static targets posed at a starting distance of 1.2 m in approximately 250 control steps. Hülse et al. [11] introduced a computational framework that integrates robotic active vision and reaching. Essential elements of this framework are sensorimotor mappings that link three different computational domains relating to visual data, gaze control and reaching. 

Some researchers have applied the combined movement of the eyes, head and body in mobile robots. In one study [12], large reorientations of the line of sight, involving combined rotations of the eyes, head, trunk and lower extremities, were executed either as fast single-step or as slow multiple-step gaze transfers. Daye et al. [13] proposed a novel approach for the control of linked systems with feedback loops for each part. The proximal parts had separate goals. In addition, an efficient and robust human tracker for a humanoid robot was implemented and experimentally evaluated in another study [14].

On the one hand, human eyes can obtain three-dimensional (3D) information from objects. This 3D information is useful for humans to make decisions. Human can shift their gaze stably and approach a target using the 3D information of the object. When the human gaze shifts to a moving target, the eyes first rotate to the target, and then the head and even the body rotate if the target leaves the sight of the eyes [15]. Therefore, the eyes, head and body move in coordination to shift the gaze to the target with minimal energy expenditure. On the other hand, when a human approaches a target, the eyes, head and body rotate to face the target and the body moves toward the target. The two movements are typically executed with the eyes, head and body acting in conjunction. A robot that can execute these two functions will be more intelligent. Such a robot would need to exploit the smooth pursuit of eyes [16], coordinated eye–head movement [17], target detection and the combined movement of the eyes, head and robot body to carry out these two functions. Studies have achieved many positive results in these aspects.

Mobile robots can track and locate objects according to 3D information. Some special cameras such as deep cameras and 3D lasers have been applied to obtain the 3D information of the environment and target. In one study [18], a nonholonomic under-actuated robot with bounded control was described that travels within a 3D region. A single sensor provided the value of an unknown scalar field at the current location of the robot. Nefti-Meziani S et al. [19] presented the implementation of a stereo-vision system integrated in a humanoid robot. The low cost of the vision system is one of the main aims, avoiding expensive investment in hardware when used in robotics for 3D perception. Namavari A et al. [20] presented an automatic system for the gauging and digitalization of 3D indoor environments. The configuration consisted of an autonomous mobile robot, a reliable 3D laser rangefinder and three elaborated software modules. 

The main forms of motion of bionic eyes include saccade [1], smooth pursuit, vergence [21], vestibule–ocular reflex (VOR) [22] and optokinetic reflex (OKR) [23]. Saccade and smooth pursuit are the two most important functions of the human eye. Saccade is used to move eyes voluntarily from one point to another by rapid jumping, while smooth pursuit can be applied to track moving targets. In addition, binocular coordination and eye–head coordination are of high importance to realize object tracking and gaze control.

It is of great significance for robots to be able change their fixation point quickly. In control models, the saccade control system should be implemented using a position servo controller to change and keep the target at the center region of the retina with minimum time consumption. Researchers have been studying the implementation of saccade on robots over the last twenty years. For example, in 1997, Bruske et al. [24] incorporated saccadic control into a binocular vision system by using the feedback error learning (FEL) strategy. In 2013, Wang et al. [25] designed an active vision system that can imitate saccade and other eye movements. The saccadic movements were implemented with an open-loop controller, which ensures faster saccadic eye movements than a closed-loop controller can accommodate. In 2015, Antonelli et al. [26] achieved saccadic movements on a robot head by using a model called recurrent architecture (RA). In this model, the cerebellum is regarded as an adaptive element used to learn an internal model, while the brainstem is regarded as a fixed-inverse model. The experimental results on the robot showed that this model is more accurate and less sensitive to the choice of the inverse model relative to the FEL model. 

The smooth pursuit system acts as a velocity servo controller to rotate eyes at the same angular rate as the target while keeping them oriented toward the desired position or in the desired region. In Robinson’s model of smooth pursuit [27], the input is the velocity of the target’s image across the retina. The velocity deviation is taken as the major stimulus to pursue and is transformed into an eye velocity command. Based on Robinson’s model, Brown [28] added a smooth predictor to accommodate time delays. Deno et al. [29] applied a dynamic neural network, which unified two apparently disparate models of smooth pursuit and dynamic element organization to the smooth pursuit system. The dynamic neural network can compensate for delays from the sensory input to the motor response. Lunghi et al. [30] introduced a neural adaptive predictor that was previously trained to accomplish smooth pursuit. This model can explain a human’s ability to compensate for the 130 ms physiological delay when they follow external targets with their eyes. Lee et al. [31] applied a bilateral OCS model on a robot head and established rudimentary prediction mechanisms for both slow and fast phases. Avni et al. [32] presented a framework for visual scanning and target tracking with a set of independent pan–tilt cameras based on model predictive control (MPC). In another study [33], the authors implemented smooth pursuit eye movement with prediction and learning in addition to solving the problem of time delays in the visual pathways. In addition, some saccade and smooth pursuit models have been validated on bionic eye systems [34,35,36,37]. Santini F et al. [34] showed that the oculomotor strategies by which humans scan visual scenes produce parallaxes that provide an accurate estimation of distance. Other studies have realized the coordinated control of eye and arm movements through configuration and training [35]. Song Y et al. [36] proposed a binocular control model, which was derived from a neural pathway, for smooth pursuit. In their smooth pursuit experiments, the maximum retinal error was less than 2.2°, which is sufficient to keep a target in the field of view accurately. An autonomous mobile manipulation system was developed in the form of a modified image-based visual servo (IBVS) controller in a study [37]. 

The above-mentioned work is significant for the development of intelligent robots. However, there are some shortcomings. First, most of the existing methods are based on the principle of neurology, and further developments and applications may be limited by people’s understanding aimed at humans. Second, only two-dimensional (2D) image information is applied when gaze shifts to targets are implemented, while 3D information is ignored. Third, the studies of smooth pursuit [16], eye–head coordination [17], gaze shift and approach are independent and have not been integrated. Fourth, bionic eyes are different from human eyes; for example, some of them are two eyes that are fixed without movement or move with only 1 DOF, whereas some of them use special cameras or a single camera. Fifth, the movements of bionic eyes and heads are performed separately, without coordination.

To overcome the shortcomings mentioned above to a certain extent, a novel control method that implements the gaze shift and approach of a robot according to 3D coordinates is proposed in this paper. A robot system equipped with bionic eyes, a head and a mobile robot is designed to help nurses deliver medicine in hospitals. In this system, both the pan and each eye have 2 DOF (namely, tilt and pan [38]), and the mobile robot can rotate and move forward over the ground. When the robot gaze shifts to the target, the 3D coordinates of the target are acquired by the bionic eyes and transferred to the eye coordination system, head coordination system and robot coordination system. The desired position of the eye, head and robot are calculated based on the 3D information of the target. Then, the eye, head and mobile robot are driven to the desired positions. When the robot approaches the target, the eye, head and mobile robot first rotate to the target and then move to the target. This method allows the robot to achieve the above-mentioned functions with minimal resource consumption and can separate the control of the eye, head and mobile robot, which can improve the interactions between robots, human beings and the environment.

The rest of the paper is organized as follows. In Section 2, the robot system platform is introduced, and the control system is presented. In Section 3, the desired position is discussed and calculated. Robot pose control is described in Section 4. The experimental results are given and discussed in Section 5; finally, conclusions are drawn in Section 6.

## 2. Platform and Control System

To study the gaze point tracking of the robot, this paper designs a robot experiment platform including the eye–head subsystem and the mobile robot subsystem.

### 2.1. Robot Platform

The physical object of the robot is shown in Figure 1. With the mobile robot as a carrier, a head with two degrees of freedom is fixed on the mobile robot, and the horizontal and vertical rotations of the head are controlled by M_hu_ and M_hd_, respectively. The bionic eye system is fixed to the head. The mobile robot is driven by two wheels, each of which is individually controlled by a servo motor. The angle and displacement of the robot platform can be determined by controlling the distance and speed of each wheel’s movement. The output shaft of each stepper motor of the head and eye is equipped with a rotary encoder to detect the position of the motor. Using the frequency multiplication technique, the resolution of the rotary encoder is 0.036°. The purpose of using a rotary encoder is to prevent the effects of lost motor motion on the 3D coordinate calculations. The movement of each motor is limited by a limit switch. The initial positioning of the eye system is based on the visual positioning plate [39].

The robot system includes two eyes and one mobile robot. To simulate the eyes and the head, six DOFs are designed in this system. The left eye’s pan and tilt are controlled by motors M_lu_ and M_ld_, respectively. The right eye’s pan and tilt are controlled by motors M_ru_ and M_rd_, respectively. The head’s pan and tilt are controlled by motors M_hu_ and M_hd_, respectively. The mobile robot has two driving wheels and can perform rotation and forward movement. When the mobile robot needs to rotate, two wheels are set to turn the same amount in different directions. When the mobile robot needs to go forward, two wheels are set to turn the same amount in the same direction.

A diagram of the robot system’s organization is shown in Figure 2. The host computer and the mobile robot motion controller, the head motion controller and the eye motion controller all communicate through the serial ports. For satisfactory communication quality and stability, the baud rate of serial communication is 9600 bps. The camera communicates with the host computer via a GigE Gigabit Network. The camera’s native resolution is 1600 × 1200 pixels. To increase the calculation speed, the system uses an image downsampled to 400 × 300 pixels.

### 2.2. Control System

Figure 3 shows the control block diagram of the gaze point tracking of the mobile robot. First, based on binocular stereo-vision perception, the binocular pose and the left and right images are used to calculate the 3D coordinates of the target [40], and the coordinates of the target in the eye coordinate system are converted to the head and mobile robot coordinate system. Then, the desired poses of the eyes, head and mobile robot are calculated according to the 3D coordinates of the target. Finally, according to the desired pose, the motor is controlled to move to the desired position, and the change in the position of the motor is converted into changes in the eyes, head and mobile robot.

The tracking and approaching motion control problem based on the target 3D coordinates [1] is equivalent to solving the index *J* minimization problem of Equation (1), where ***f****_i_* is the current state vector of the joint pose of the eye, head and mobile robot and ***f***_q_ is the desired state vector:(1)$$J=\left\|f_{i}-f_{q}\right\|$$
where *J* is the indicator function.

Figure 4a shows the definition of each coordinate system of the robot. The coordinate system of the eye is *O*_e_*X*_e_*Y*_e_*Z*_e_, which coincides with the left motion module’s base coordinate system at the initial position. The head coordinate system is *O*_h_*X*_h_*Y*_h_*Z*_h_, and the coordinates ***P***_h_ (*x*_h_, *y*_h_, *z*_h_) of the point ***P*** in the head coordinate system can be calculated using the coordinates ***P***_e_ (*x*_e_, *y*_e_, *z*_e_) in the eye coordinate system. The definitions of *d_x_* and *d_y_* are shown in Figure 4b. The robot coordinate system *O*_w_*X*_w_*Y*_w_*Z*_w_ coincides with the head coordinate system of the initial position. In the bionic eye system, the axis of rotation of the robot approximately coincides with *Y*_w_.

Figure 4b,c show the definition of each system parameter. ^l^*θ*_p_ and ^l^*θ*_t_ are the pan and tilt of the left eye, respectively. ^r^*θ*_p_ and ^r^*θ*_t_ are the pan and tilt of the right eye, respectively. ^h^*θ*_p_ and ^h^*θ*_t_ are the pan and tilt of the head, respectively. The angle of the robot that rotates around the *Y*_w_ axis is ^w^*θ*_p_. The robot can not only rotate around *Y*_w_ but can also shift in the *X*_w_*O*_w_*Z*_w_ plane. When the robot moves, the robot coordinate system at time *i* is the base coordinate system, and the position of the robot at time *i* + 1 relative to the base coordinate system is ^w^***P***_m_ (^w^*x*_m_, ^w^*z*_m_). When the robot performs gaze point tracking or approaches the target, the 3D coordinates of the target are first calculated at time *i*, and then the desired posture ***f***_q_ of each part of the robot at time *i* + 1 is calculated according to the 3D coordinates of the target. When the current pose ***f****_i_* of the robot system is equal to the desired pose, the robot maintains the current pose; when not equal, the system controls the various parts of the robot to move to the desired pose. The current pose vector of the robot system is ***f****_i_* = (^w^*x*_m*i*_, ^w^*z*_m*i*_, ^w^*θ*_p*i*_, ^h^*θ*_p*i*_, ^h^*θ*_t*i*_, ^l^*θ*_p*i*_, ^l^*θ*_t*i*_, ^r^*θ*_p*i*_, ^r^*θ*_t*i*_), and the desired pose is ***f***_q_ = (^w^*x*_mq_, ^w^*z*_mq_, ^w^*θ*_pq_, ^h^*θ*_pq_, ^h^*θ*_tq_, ^l^*θ*_pq_, ^l^*θ*_tq_, ^r^*θ*_pq_, ^r^*θ*_tq_). When performing in situ gaze point tracking, the robot performs only pure rotation and does not move forward. When the robot approaches the target, it first turns to the target and then moves straight toward the target. Therefore, the definition of ***f***_q_ in the two tasks is different. Let ^g^***f***_q_ be the desired pose when the gaze point is tracked and ^a^***f***_q_ be the desired pose of the robot when approaching the target. 

After analyzing the control system, we found that the most important step in solving this control problem is to determine the desired pose.

## 3. Desired Pose Calculation

When performing in situ gaze point tracking, the robot performs only pure rotation and does not move forward. When the robot approaches the target, it first turns to the target and then moves straight toward the target. Therefore, the calculation of the desired pose can be divided into two sub-problems: (1) desired pose calculation for in situ gaze point tracking and (2) desired pose calculation for approaching gaze point tracking.

The optimal observation position is used for the accurate acquisition of 3D coordinates. The 3D coordinate accuracy is related to the baseline, time difference and image distortion. In the bionic eye platform, the baseline is changed with the changes in the cameras’ positions because the optical center is not coincident with the center of rotation. The 3D coordinate error of the target is smaller when the baseline of the two cameras is longer. Therefore, it is necessary to keep the baseline unchanged. On the other hand, there is a time difference caused by unstick synchronization between image acquisition and camera position acquisition. In addition, it is necessary to keep the target in the center areas of the two camera images to obtain accurate 3D coordinates of the target.

### 3.1. Optimal Observation Position of Eyes

In the desired pose of the robot, the most important aspect is the expected pose of the bionic eye [40]. Following the definition of this parameter, the calculation of the desired pose of the robot system is greatly simplified; thus, we present an engineering definition here of the desired pose of the bionic eye. 

As shown in Figure 5, ^l^***m****_i_* (^l^*u_i_*, ^l^*v_i_*) and ^r^(^r^*u_i_*, ^r^*v_i_*) are the image coordinates of point ^e^***P*** in the camera at time *i*. ^l^***m***_o_ and ^r^***m***_o_ are the image centers of the left and right cameras, respectively. ^l^***P*** is the vertical point of ^e^***P*** along the line ^l^*O*_c_^l^*Z*_c_, and ^r^***P*** is the vertical point of ^e^***P*** along the line ^r^*O*_c_^r^*Z*_c_. ^l^∆*m* is the distance between ^l^***m*** and ^l^***m***_o_. ^r^∆*m* is the distance between ^r^***m*** and ^r^***m***_o_. *D*_b_ is the baseline length. The pan angles of the left and right cameras in the optimal observation position are ^l^*θ*_p_ and ^r^*θ*_p_, respectively. The tilt angles of the left and right cameras in the optimal observation position are ^l^*θ*_t_ and ^r^*θ*_t_, respectively. ***P***_ob_ (^l^*θ*_p_, ^l^*θ*_t_, ^r^*θ*_p_, ^r^*θ*_t_) is the optimal observation position.

When the two eyeballs of the bionic eye move relative to each other, the 3D coordinates of the target obtained by the bionic eye produces a large error. To characterize this error, we give a detailed analysis of its origins in Appendix A. Through analysis, we obtain the following conclusions to reduce the measurement error of the bionic eye:

(1) Make the length of *D*_b_ long enough, and maintain as much length as possible during the movement;

(2) Try to observe the target closer to the target so that the depth error is as small as possible;

(3) During the movement of the bionic eye, control the two cameras so that they move at the same angular velocity;

(4) Try to keep the target symmetrical, and make ^l^Δ*m* and ^r^Δ*m* as equal as possible in the left and right camera images.

Based on these four methods, the motion strategy of the motor is designed, and the measurement accuracy of the target’s 3D information can be effectively improved.

According to the conclusion, we can define a definition of the optimal observed pose of the bionic eye to reduce the measurement error.

The optimal observation position needed to meet the conditions is listed in Equation (2). When the target is very close to the eyes, the target’s optimal observation position cannot be obtained because the image position of the target can be kept at the image center region. It is challenging to obtain the optimal solution of the observation position based on Equation (12). However, a suboptimal solution can be obtained by using a simplified calculation method. First, ^l^*θ*_t_ and ^r^*θ*_t_ are calculated in the case that ^l^*θ*_t_ and ^r^*θ*_t_ are equal to zero; then, ^l^*θ*_t_ and ^r^*θ*_t_ are calculated while ^l^*θ*_t_ and ^r^*θ*_t_ are kept equal to the calculated value. Trial-and-error methods can be used to obtain the optimal solution when the suboptimal solution is obtained.
(2)$$\left\{\begin{array}{l} {}^{l}\theta_{\mathrm{pq}}={}^{r}\theta_{\mathrm{pq}}=\theta_{p}\\ {}^{l}\theta_{\mathrm{tq}}={}^{r}\theta_{\mathrm{tq}}=\theta_{t}\\ {}^{l}\Delta m={-}^{r}\Delta m \end{array}\right.$$
where
(3)$${}^{l}\Delta m=\left(\begin{array}{l} {}^{l}\Delta u\\ {}^{l}\Delta v \end{array}\right)=\left(\begin{array}{c} {}^{l}u_{i}-{}^{l}u_{0}\\ {}^{l}v_{i}-{}^{l}v_{0} \end{array}\right)$$
(4)$${}^{r}\Delta m=\left(\begin{array}{l} {}^{r}\Delta u\\ {}^{r}\Delta v \end{array}\right)=\left(\begin{array}{c} {}^{r}u_{i}-{}^{r}u_{0}\\ {}^{r}v_{i}-{}^{r}v_{0} \end{array}\right)$$

### 3.2. Desired Pose Calculation for In Situ Gaze Point Tracking

When the range of target motion is large and the desired posture of the eyeball exceeds its reachable posture, the head and mobile robot move to keep the target in the center region of the image. In robotic systems, eye movements tend to consume the least amount of resources and do not have much impact on the stability of the head and mobile robot during exercise. Head rotation consumes more resources than the eyeball but consumes fewer resources than trunk rotation. At the same time, the rotation of the head affects the stability of the eyeball but does not have much impact on the stability of the trunk. Mobile robot rotation consumes the most resources and has a large impact on the stability of the head and eyeball. When tracking the target, one needs only to keep the target in the center region of the binocular image. Therefore, when performing gaze point tracking, the movement mechanism of the head, eyes and mobile robot are designed with the principle of minimal resource consumption and maximum system stability. When the eyeball can perceive the 3D coordinates of the target in the reachable and optimal viewing posture, only the eye is rotated; otherwise, the head is rotated. The head also has an attainable range of poses. When the desired pose exceeds this range, the mobile robot needs to be turned so that the bionic eye always perceives the 3D coordinates of the target in the optimal viewing position. Let ^h^*γ*_p_ and ^h^*γ*_t_ be the angles between the head and the gaze point in the *X*_h_*O*_h_*Z*_h_ and *Y*_h_*O*_h_*Z*_h_ planes, respectively. The range of binocular rotation in the horizontal direction is [−^e^*θ*_pmax_, ^e^*θ*_pmax_], and the range of binocular rotation in the vertical direction is [−^e^*θ*_tmax_, ^e^*θ*_tmax_]. The range of head rotation in the horizontal direction is [−^h^*θ*_pmax_, ^h^*θ*_pmax_], and the range of head rotation in the vertical direction is [−^h^*θ*_tmax_, ^h^*θ*_tmax_]. For the convenience of calculation, the angles between the head and the fixation point in the horizontal direction and the vertical direction are designated as [−^h^*γ*_pmax_, ^h^*γ*_pmax_] and [−^h^*γ*_tmax_, ^h^*γ*_tmax_], respectively. When the angle between the head and the target exceeds a set threshold, the head needs to be rotated to the ${}^{h}\theta_{p}^{\prime }$ and ${}^{h}\theta_{t}^{\prime }$ positions in the horizontal and vertical directions, respectively. When ${}^{h}\theta_{p}^{\prime }$ exceeds the angle that the head can attain, the angle at which the mobile robot needs to be compensated is ^w^*θ*_p_. In the in situ gaze point tracking task, the cart does not need to translate in the *X*_w_*O*_w_*Z*_w_ plane, so *x*_w_ = 0, and *z*_w_ = 0. Furthermore, according to the definition of the optimal observation pose of the bionic eye, the conditions that ^g^***f***_q_ should satisfy are
(5)$${}^{g}f_{q}=\left(\begin{array}{l} {}^{w}x_{\mathrm{mq}}=0\\ {}^{w}z_{\mathrm{mq}}=0\\ {}^{w}\theta_{\mathrm{pq}}=\{\theta \left|\right.\;\;|\theta |\leq 2\pi ,{}^{h}\theta_{\mathrm{pq}}+\theta ={}^{h}\theta_{p}^{\prime }\}\\ {}^{h}\theta_{\mathrm{pq}}=\{\theta \left|\right.\;\;|\theta |\leq {}^{h}\theta_{\mathrm{pmax}},{|}^{h}\gamma_{p}|\leq^{h}\gamma_{\mathrm{pmax}}\}\\ {}^{h}\theta_{\mathrm{tq}}=\{\theta \left|\right.\;\;|\theta |\leq {}^{h}\theta_{\mathrm{tmax}},{|}^{h}\gamma_{t}|\leq^{h}\gamma_{\mathrm{tmax}}\}\\ {}^{l}\theta_{\mathrm{pq}}={}^{r}\theta_{\mathrm{pq}}=\{\theta \left|\right.\;\;|\theta |\leq {}^{e}\theta_{\mathrm{pmax}},{}^{l}{\Delta m}_{l}=-\Delta m_{r}\}\\ {}^{l}\theta_{\mathrm{tq}}={}^{r}\theta_{\mathrm{tq}}=\{\theta \left|\right.\;\;|\theta |\leq {}^{e}\theta_{\mathrm{tmax}},\Delta m_{l}=-\Delta m_{r}\} \end{array}\right)$$

The desired pose needs to be calculated based on the 3D coordinates of the target. Therefore, to obtain the desired pose, it is necessary to acquire the 3D coordinates of the target according to the current pose of the robot.

#### 3.2.1. Three-Dimensional Coordinate Calculation

The mechanical structure and coordinate settings of the system are shown in Figure 6a. The principle of binocular stereoscopic 3D perception is shown in Figure 6b. E is the eye coordinate system, E_l_ is the left motion module’s end coordinate system, E_r_ is the right motion module’s end coordinate system, B_l_ is the left motion module’s base coordinate system, B_r_ is the right motion module’s base coordinate system, C_l_ is the left camera coordinate system and C_r_ is the right camera coordinate system. In the initial position, E_l_ coincides with B_l_, and E_r_ overlaps with B_r_. When the binocular system moves, the base coordinate system does not change. ^l^***T*** represents the transformation matrix of the eye coordinate system E to the left motion module’s base coordinate system B_l_, ^r^***T*** represents the transformation matrix of E to B_r_, ^l^***T***_e_ represents the transformation matrix of B_l_ to E_l_, ^r^***T***_e_ represents the transformation matrix of B_r_ to E_r_ and ^l^***T***_m_ represents the leftward motion. The module end coordinate system corresponds to the transformation matrix of the left camera coordinate system, and ^r^***T***_m_ represents the transformation matrix of the right motion module’s end coordinate system to the right camera coordinate system. ^l^***T***_r_ represents the transformation matrix of the right camera coordinate system to the left camera coordinate system at the initial position.

The origin ^l^*O*_c_ of C_l_ lies at the optical center of the left camera, the ^l^*Z*_c_ axis points in the direction of the object parallel to the optical axis of the camera, the ^l^*X*_c_ axis points horizontally to the right along the image plane and the ^l^*Y*_c_ axis points vertically downward along the image plane. The origin ^r^*O*_c_ of C_r_ lies at the optical center of the right camera, ^r^*Z*_c_ is aligned with the direction of the object parallel to the optical axis of the camera, ^r^*X*_c_ points horizontally to the right along the image plane and ^r^*Y*_c_ points vertically downward along the image plane. E_l_’s origin ^l^*O*_e_ is set at the intersection of the two rotation axes of the left motion module, ^l^*Z*_e_ is perpendicular to the two rotation axes and points to the front of the platform, ^l^*X*_e_ coincides with the vertical rotation axis and ^l^*Y*_e_ coincides with the horizontal rotation axis. Similarly, the origin ^r^*O*_e_ of the coordinate system E_r_ is set at the intersection of the two rotation axes of the right motion module, ^r^*Z*_e_ is perpendicular to the two rotation axes and points toward the front of the platform, ^r^*X*_e_ coincides with the vertical rotation axis and ^r^*Y*_e_ coincides with the horizontal rotation axis.

The left motion module’s base coordinates system B_l_ coincides with the eye coordinate system E; thus, ^l^***T*** consists of an identity matrix. To calculate the 3D coordinates of the feature points in real time from the camera pose, it is necessary to calculate ^r^***T***. At the initial position of the system, the external parameters ^l^***T***_r_ of the left and right cameras are calibrated offline, as are the hand–eye parameters of the left–right motion module to the camera coordinate system.

When the system is in its initial configuration, the coordinates of point ***P*** in the eye coordinate system are ***P***_e_ (*x*_e_, *y*_e_, *z*_e_). Its coordinates in B_l_ are ^l^***P***_e_ (^l^*x*_e_, ^l^*y*_e_, ^l^*z*_e_), and its coordinates ^l^***P***_c_ (^l^*x*_c_, ^l^*y*_c_, ^l^*z*_c_) in C_l_ are
(6)$${}^{l}P_{c}={}^{l}T_{m}^{-1}P_{e}$$

The coordinates ^r^***P***_e_ (^r^*x*_e_, ^r^*y*_e_, ^r^*z*_e_) of point *P* in B_r_ are
(7)$${}^{r}P_{e}={}^{r}TP_{e}$$

The coordinates ^r^***P***_c_ (^r^*x*_c_, ^r^*y*_c_, ^r^*z*_c_) of point ***P*** in C_r_ are
(8)$${}^{r}P_{c}={}^{r}T_{m}^{-1}{}^{r}TP_{e}$$

The point in C_r_ is transformed into C_l_:(9)$${}^{l}P_{c}={}^{l}T_{r}{}^{r}T_{m}^{-1}{}^{r}TP_{e}$$

Based on the Equations (6) and (9), ^r^***T*** is available:(10)$${}^{r}T={}^{r}T{}_{m}{}^{l}T_{r}^{-1}{}^{l}T_{m}^{-1}$$

During the movement of the system, when the left motion module rotates by ^l^*θ*_p_ and ^l^*θ*_t_ in the horizontal and vertical directions, respectively, the transformation relationship between B_l_ and E_l_ is
(11)$${}^{l}T_{e}=\left(\begin{array}{cc} \mathrm{Rot}(Y,{}^{l}\theta_{p})\mathrm{Rot}(X,{}^{l}\theta_{t}) & 0\\ 0 & 1 \end{array}\right)$$

The coordinates of point ***P*** in C_l_ are
(12)$${}^{l}P_{e}={}^{l}T_{m}^{-1}{}^{l}T_{e}P_{w}={}^{l}T_{d}P_{e}$$

Assume that
(13)$${}^{l}T_{d}=\left(\begin{array}{cccc} {}^{l}n_{x} & {}^{l}o_{x} & {}^{l}a_{x} & {}^{l}p_{x}\\ {}^{l}n_{y} & {}^{l}o_{y} & {}^{l}a_{y} & {}^{l}p_{y}\\ {}^{l}n_{z} & {}^{l}o_{z} & {}^{l}a_{z} & {}^{l}p_{z}\\ 0 & 0 & 0 & 1 \end{array}\right)$$

The point ^l^***P***_1c_ (^l^*x*_1c_, ^l^*y*_1c_) at which line *P*^l^***O***_c_ intersects ^l^*Z*_c_ = 1 is
(14)$$\left(\begin{array}{c} {}^{l}x_{1c}\\ {}^{l}y_{1c} \end{array}\right)=\left(\begin{array}{c} \frac{{}^{l}n_{x}x_{e}+{}^{l}o_{x}y_{e}+{}^{l}a_{x}z_{e}+{}^{l}p_{x}}{{}^{l}n_{z}x_{e}+{}^{l}o_{z}y_{e}+{}^{l}a_{z}z_{e}+{}^{l}p_{z}}\\ \frac{{}^{l}n_{y}x_{e}+{}^{l}o_{y}y_{e}+{}^{l}a_{y}z_{e}+{}^{l}p_{y}}{{}^{l}n_{z}x_{e}+{}^{l}o_{z}y_{e}+{}^{l}a_{z}z_{e}+{}^{l}p_{z}} \end{array}\right)$$

The image coordinates of ^l^***P***_1c_ in the left camera are ***m***_l_ (*u*_l_, *v*_l_), (^l^*x*_1c_, ^l^*y*_1c_) and (*u*_l_, *v*_l_) and can be converted by the parameters of the camera. According to the camera’s internal parameter model, the following can be obtained:(15)$$\left(\begin{array}{c} {}^{l}x_{1c}\\ {}^{l}y_{1c}\\ 1 \end{array}\right)={}^{l}M_{\mathrm{in}}^{-1}\left(\begin{array}{c} u_{l}\\ v_{l}\\ 1 \end{array}\right)$$
where ^l^***M***_in_ is the internal parameter matrix of the left camera. The value of (^l^*x*_1c_, ^l^*y*_1c_) can be obtained by the image coordinates of ^l^***P***_1c_, and the parameters of the left camera can be obtained by substituting (15) into (14):(16)$$\left\{\begin{array}{l} ({}^{l}n_{x}-{}^{l}x_{1c}{}^{l}n_{z})x_{e}+({}^{l}o_{x}-{}^{l}x_{1c}{}^{l}o_{z})y_{e}+({}^{l}a_{x}-{}^{l}x_{1c}{}^{l}a_{z})z_{e}+{}^{l}p_{x}-{}^{l}x_{1c}{}^{l}p_{z}=0\\ ({}^{l}n_{y}-{}^{l}y_{1c}{}^{l}n_{z})x_{e}+({}^{l}o_{y}-{}^{l}y_{1c}{}^{l}o_{z})y_{e}+({}^{l}a_{y}-{}^{l}y_{1c}{}^{l}a_{z})z_{e}+{}^{l}p_{y}-{}^{l}y_{1c}{}^{l}p_{z}=0 \end{array}\right.$$

During the motion of the system, when the right motion module rotates through ^r^*θ*_p_ and ^r^*θ*_t_ in the horizontal and vertical directions, respectively, the transformation relationship between B_r_ and E_r_ is
(17)$${}^{r}T{}_{e}=\left(\begin{array}{cc} \mathrm{Rot}(Y,{}^{r}\theta_{p})\mathrm{Rot}(X,{}^{r}\theta_{t}) & 0\\ 0 & 1 \end{array}\right)$$

The coordinates of point ***P*** in C_r_ are
(18)$${}^{r}P_{e}={}^{r}T_{m}^{-1}{}^{r}T{}_{e}{}^{r}TP_{e}={}^{r}T{}_{d}P_{e}$$

Assume that
(19)$${}^{r}T{}_{d}=\left(\begin{array}{cccc} {}^{r}n_{x} & {}^{r}o_{x} & {}^{r}a_{x} & {}^{r}p_{x}\\ {}^{r}n_{y} & {}^{r}o_{y} & {}^{r}a_{y} & {}^{r}p_{y}\\ {}^{r}n_{z} & {}^{r}o_{z} & {}^{r}a_{z} & {}^{r}p_{z}\\ 0 & 0 & 0 & 1 \end{array}\right)$$

The point ^l^***P***_1c_ (^r^*x*_1c_, ^r^*y*_1c_) at which line *P*^r^***O***_c_ intersects ^r^*Z*_c_ = 1 is
(20)$$\left(\begin{array}{c} {}^{r}x_{1c}\\ {}^{r}y_{1c} \end{array}\right)=\left(\begin{array}{c} \frac{{}^{r}n_{x}x_{e}+{}^{r}o_{x}y_{e}+{}^{r}a_{x}z_{e}+{}^{r}p_{x}}{{}^{r}n_{z}x_{e}+{}^{r}o_{z}y_{e}+{}^{r}a_{z}z_{e}+{}^{r}p_{z}}\\ \frac{{}^{r}n_{y}x_{e}+{}^{r}o_{y}y_{e}+{}^{r}a_{y}z_{e}+{}^{r}p_{y}}{{}^{r}n_{z}x_{e}+{}^{r}o_{z}y_{e}+{}^{r}a_{z}z_{e}+{}^{r}p_{z}} \end{array}\right)$$

The image coordinates of ^r^***P***_1c_ in the camera, namely, ***m***_r_ (*u*_r_, *v*_r_), (^r^*x*_1c_, ^r^*y*_1c_) and (*u*_r_, *v*_r_), can be converted using the parameters of the camera. According to the camera’s internal parameter model, the following can be obtained:(21)$$\left(\begin{array}{c} {}^{r}x_{1c}\\ {}^{r}y_{1c}\\ 1 \end{array}\right)={}^{l}M_{\mathrm{in}}^{-1}\left(\begin{array}{c} u_{r}\\ v_{r}\\ 1 \end{array}\right)$$
where ^r^***M***_in_ is the inner parameter matrix of the right camera. The value of (^r^*x*_1c_, ^r^*y*_1c_) can be obtained by the image coordinates of ^r^***P***_1c_ and the parameters in the camera, and the following can be obtained by substituting (21) into (20):(22)$$\left\{\begin{array}{l} ({}^{r}n_{x}-{}^{r}x_{1c}{}^{r}n_{z})x_{e}+({}^{r}o_{x}-{}^{r}x_{1c}{}^{r}o_{z})y_{e}+({}^{r}a_{x}-{}^{r}x_{1c}{}^{r}a_{z})z_{e}+{}^{r}p_{x}-{}^{r}x_{1c}{}^{r}p_{z}=0\\ ({}^{r}n_{y}-{}^{r}y_{1c}{}^{r}n_{z})x_{e}+({}^{r}o_{y}-{}^{r}y_{1c}{}^{r}o_{z})y_{e}+({}^{r}a_{y}-{}^{r}y_{1c}{}^{r}a_{z})z_{e}+{}^{r}p_{y}-{}^{r}y_{1c}{}^{r}p_{z}=0 \end{array}\right.$$

Four equations can be obtained from Equations (16) and (22) for *x*_e_, *y*_e_ and *z*_e_, and the 3D coordinates of point ***P***_e_ can be calculated by the least squares method.

The 3D coordinates ***P***_h_ (*x*_h_, *y*_h_, *z*_h_) in the head coordinate system can be obtained by Equation (23). *d*_x_ and *d*_y_ are illustrated in Figure 4.
(23)$$\left(\begin{array}{c} x_{h}\\ y_{h}\\ z_{h} \end{array}\right)=\left(\begin{array}{c} x_{e}-d_{x}\\ y_{e}-d_{y}\\ z_{e} \end{array}\right)$$

Let the angles at which the current moment of the head rotate relative to the initial position be ^h^*θ*_p*i*_ and ^h^*θ*_t*i*_; the coordinates of the target in the robot coordinate system are
(24)$$\left(\begin{array}{c} \begin{array}{c} {}^{w}x_{m}\\ {}^{w}y_{m}\\ {}^{w}z_{m} \end{array}\\ 1 \end{array}\right)={\left(\begin{array}{cc} \mathrm{Rot}(X{,}^{h}\theta_{ti})\mathrm{Rot}(Y{,}^{h}\theta_{pi}) & 0\\ 0 & 1 \end{array}\right)}^{-1}\left(\begin{array}{c} \begin{array}{c} x_{h}\\ y_{h}\\ z_{h} \end{array}\\ 1 \end{array}\right)$$

According to the 3D coordinates of the target in the head coordinate system, the angle between the target and *Z*_h_ in the horizontal direction and the vertical direction can be obtained as follows: (25)$${}^{h}\gamma_{p}=\arctan(\frac{x_{h}}{z_{h}})$$
(26)$${}^{h}\gamma_{t}=\arctan(\frac{y_{h}}{z_{h}})$$

When ^h^*γ*_p_ and ^h^*γ*_t_ exceed a set threshold, the head needs to rotate. To leave a certain margin for the rotation of the eyeball and for the convenience of calculation, the angles required for the head to rotate in the horizontal direction and the vertical direction are calculated by the principle shown in Figure 7a,b, respectively. Figure 7a shows the calculation principle of the horizontal direction angle when the target’s *x* coordinates of the head coordinate system is greater than zero. After the head is rotated to ${}^{h}\theta_{p}^{\prime }$, the target point is on the ^l^*Z*_e_ axis of the left motion module end coordinate system, and the left motion module reaches the maximum rotatable threshold ^e^*θ*_pmax_. Figure 7b shows the calculation principle of the vertical direction when the target’s *y* coordinates of the head coordinate system are greater than *d_y_*. After the head is rotated to ${}^{h}\theta_{t}^{\prime }$, the target point is on the *Z*_e_ axis of the eye coordinate system, and the eye reaches the maximum threshold ^e^*θ*_tmax_ that can be rotated. 

#### 3.2.2. Horizontal Rotation Angle Calculation

Let the current angle of the head in the horizontal direction be ^h^*θ*_p*i*_. When the head is rotated in the horizontal direction to ${}^{h}\theta_{p}^{\prime }$, the 3D coordinates of the target in the new head coordinate system are
(27)$$\left(\begin{array}{c} \begin{array}{c} x_{h}^{\prime }\\ y_{h}^{\prime }\\ z_{h}^{\prime } \end{array}\\ 1 \end{array}\right)=\left(\begin{array}{cccc} \cos{(}^{h}\theta_{p}^{\prime }{-}^{h}\theta_{pi}) & 0 & -\sin{(}^{h}\theta_{p}^{\prime }{-}^{h}\theta_{pi}) & 0\\ 0 & 1 & 0 & 0\\ \sin{(}^{h}\theta_{p}^{\prime }{-}^{h}\theta_{pi}) & 0 & \cos{(}^{h}\theta_{p}^{\prime }{-}^{h}\theta_{pi}) & 0\\ 0 & 0 & 0 & 1 \end{array}\right)\left(\begin{array}{c} \begin{array}{c} x_{h}\\ y_{h}\\ z_{h} \end{array}\\ 1 \end{array}\right)$$

Therefore,
(28)$$\left(\begin{array}{c} x_{h}^{\prime }\\ y_{h}^{\prime }\\ z_{h}^{\prime } \end{array}\right)=\left(\begin{array}{c} x_{h}\cos{(}^{h}\theta_{p}^{\prime }{-}^{h}\theta_{pi})-z_{h}\sin{(}^{h}\theta_{p}^{\prime }{-}^{h}\theta_{pi})\\ y_{h}\\ x_{h}\sin{(}^{h}\theta_{p}^{\prime }{-}^{h}\theta_{pi})+z_{h}\cos{(}^{h}\theta_{p}^{\prime }{-}^{h}\theta_{pi}) \end{array}\right)$$

The coordinates of the target in the new eye coordinate system are
(29)$$\left(\begin{array}{c} {}^{e}x_{h}^{\prime }\\ {}^{e}y_{h}^{\prime }\\ {}^{e}z_{h}^{\prime } \end{array}\right)=\left(\begin{array}{c} d_{x}+x_{h}\cos{(}^{h}\theta_{p}^{\prime }{-}^{h}\theta_{pi})-z_{h}\sin{(}^{h}\theta_{p}^{\prime }{-}^{h}\theta_{pi})\\ y_{h}+d_{y}\\ x_{h}\sin{(}^{h}\theta_{p}^{\prime }{-}^{h}\theta_{pi})+z_{h}\cos{(}^{h}\theta_{p}^{\prime }{-}^{h}\theta_{pi}) \end{array}\right)$$

After turning, the left motion module reaches the maximum threshold ^e^*θ*_pmax_ that can be rotated, so that
(30)$$\tan{(}^{e}\theta_{\mathrm{pmax}})=\frac{{}^{e}z_{h}^{\prime }}{{}^{e}x_{h}^{\prime }}=\frac{x_{h}\sin{(}^{h}\theta_{p}^{\prime }{-}^{h}\theta_{pi})+z_{h}\cos{(}^{h}\theta_{p}^{\prime }{-}^{h}\theta_{pi})}{d_{x}+x_{h}\cos{(}^{h}\theta_{p}^{\prime }{-}^{h}\theta_{pi})-z_{h}\sin{(}^{h}\theta_{p}^{\prime }{-}^{h}\theta_{pi})}$$

Simplifying Equation (30), we have
(31)$$\sin{(}^{h}\theta_{p}^{\prime }{-}^{h}\theta_{pi})=\frac{d_{x}\tan{(}^{e}\theta_{\mathrm{qmax}})}{x_{h}+z_{h}\tan{(}^{e}\theta_{\mathrm{qmax}})}+\frac{x_{h}\tan{(}^{e}\theta_{\mathrm{qmax}})-z_{h}}{x_{h}+z_{h}\tan{(}^{e}\theta_{\mathrm{qmax}})}\cos{(}^{h}\theta_{p}^{\prime }{-}^{h}\theta_{pi})$$

Assume that
(32)$$\left\{\begin{array}{l} k_{1}=\frac{x_{h}\tan{(}^{e}\theta_{\mathrm{qmax}})-z_{h}}{x_{h}+z_{h}\tan{(}^{e}\theta_{\mathrm{qmax}})}\\ k_{2}=\frac{d_{x}\tan{(}^{e}\theta_{\mathrm{qmax}})}{x_{h}+z_{h}\tan{(}^{e}\theta_{\mathrm{qmax}})} \end{array}\right.$$

According to the triangular relationship,
(33)$${\left[k_{1}\cos{(}^{h}\theta_{p}^{\prime }{-}^{h}\theta_{pi})+k_{2}\right]}^{2}+\cos^{2}{(}^{h}\theta_{p}^{\prime }{-}^{h}\theta_{pi})=1$$

The solution of Equation (33) is
(34)$$\cos{(}^{h}\theta_{p}^{\prime }{-}^{h}\theta_{pi})=\frac{-k_{1}k_{2}\pm \sqrt{k_{1}^{2}-k_{2}^{2}+1}}{k_{1}^{2}+1}$$

Therefore,
(35)$${}^{h}\theta_{p}^{\prime }{=}^{h}\theta_{pi}+\arccos(\frac{-k_{1}k_{2}\pm \sqrt{k_{1}^{2}-k_{2}^{2}+1}}{k_{1}^{2}+1})$$

Equation (35) has two solutions; therefore, we choose the solution in which the deviation *e* of Equation (36) is minimized:(36)$$e=\left|\tan{(}^{e}\theta_{\mathrm{qmax}})-\frac{x_{h}\sin{(}^{h}\theta_{p}^{\prime }{-}^{h}\theta_{pi})+z_{h}\cos{(}^{h}\theta_{p}^{\prime }{-}^{h}\theta_{pi})}{d_{x}+x_{h}\cos{(}^{h}\theta_{p}^{\prime }{-}^{h}\theta_{pi})-z_{h}\sin{(}^{h}\theta_{p}^{\prime }{-}^{h}\theta_{pi})}\right|$$

When the obtained ^h^$\theta_{p}^{\prime }$ is outside of the range [−^h^*θ*_pmax_, ^h^*θ*_pmax_], the value of ^h^*θ*_pq_ is
(37)$${}^{h}\theta_{\mathrm{pq}}=\left\{\begin{array}{lll} {}^{h}\theta_{\mathrm{pmax}} & , & {}^{h}\theta_{p}^{\prime }\geq^{h}\theta_{\mathrm{pmax}}\\ {-}^{h}\theta_{\mathrm{pmax}} & , & {}^{h}\theta_{p}^{\prime }\leq {-}^{h}\theta_{\mathrm{pmax}}\\ {}^{h}\theta_{p}^{\prime } & , & \mathrm{else} \end{array}\right.$$

Finally, one can obtain the ^w^*θ*_pq_ value:(38)$${}^{w}\theta_{\mathrm{pq}}=\left\{\begin{array}{lll} {}^{h}\theta_{p}^{\prime }{-}^{h}\theta_{\mathrm{pmax}} & , & {}^{h}\theta_{p}^{\prime }{>}^{h}\theta_{\mathrm{pmax}}\\ {}^{h}\theta_{p}^{\prime }{+}^{h}\theta_{\mathrm{pmax}} & , & {}^{h}\theta_{p}^{\prime }<{-}^{h}\theta_{p\max}\\ 0 & , & \mathrm{else} \end{array}\right.$$

Based on the same principle, when the *x* coordinate of the target in the head coordinate system is less than 0, the coordinates of the target in the right motion module base coordinate system after the rotation are
(39)$$\left(\begin{array}{c} {}^{r}x_{e}^{\prime }\\ {}^{r}y_{e}^{\prime }\\ {}^{r}z_{e}^{\prime }\\ 1 \end{array}\right)=\left(\begin{array}{c} x_{h}\cos{(}^{h}\theta_{p}^{\prime }{-}^{h}\theta_{pi})-z_{h}\sin{(}^{h}\theta_{p}^{\prime }{-}^{h}\theta_{pi})-d_{x}\\ y_{h}+d_{y}\\ x_{h}\sin{(}^{h}\theta_{p}^{\prime }{-}^{h}\theta_{pi})+z_{h}\cos{(}^{h}\theta_{p}^{\prime }{-}^{h}\theta_{pi})\\ 1 \end{array}\right)$$

After turning, the right motion module reaches −^e^*θ*_pmax_, and the following can be obtained:(40)$$\tan({-}^{e}\theta_{\mathrm{qmax}})=\frac{{}^{r}z_{e}^{\prime }}{{}^{r}x_{e}^{\prime }}=\frac{x_{h}\sin{(}^{h}\theta_{p}^{\prime }{-}^{h}\theta_{pi})+z_{h}\cos{(}^{h}\theta_{p}^{\prime }{-}^{h}\theta_{pi})}{x_{h}\cos{(}^{h}\theta_{p}^{\prime }{-}^{h}\theta_{pi})-z_{h}\sin{(}^{h}\theta_{p}^{\prime }{-}^{h}\theta_{pi})-d_{x}}$$

We simplify Equation (40) as follows:(41)$$\sin{(}^{h}\theta_{p}^{\prime }{-}^{h}\theta_{pi})=\frac{d_{x}\tan{(}^{e}\theta_{\mathrm{qmax}})}{x_{h}-z_{h}\tan{(}^{e}\theta_{\mathrm{qmax}})}-\frac{x_{h}\tan{(}^{e}\theta_{\mathrm{qmax}})+z_{h}}{x_{h}-z_{h}\tan{(}^{e}\theta_{\mathrm{qmax}})}\cos{(}^{h}\theta_{p}^{\prime }{-}^{h}\theta_{pi})$$

Let
(42)$$\left\{\begin{array}{l} k_{1}^{\prime }=-\frac{x_{h}\tan{(}^{e}\theta_{\mathrm{qmax}})+z_{h}}{x_{h}-z_{h}\tan{(}^{e}\theta_{\mathrm{qmax}})}\\ k_{2}^{\prime }=\frac{d_{x}\tan{(}^{e}\theta_{\mathrm{qmax}})}{x_{h}-z_{h}\tan{(}^{e}\theta_{\mathrm{qmax}})} \end{array}\right.$$

The same two solutions are available:(43)$${}^{h}\theta_{p}^{\prime }{=}^{h}\theta_{pi}+\arccos(\frac{-k_{1}^{\prime }k_{2}^{\prime }\pm \sqrt{{\left(k_{1}^{\prime }\right)}^{2}-{\left(k_{2}^{\prime }\right)}^{2}+1}}{{\left(k_{1}^{\prime }\right)}^{2}+1})$$

Select the solution in which the deviation *e* of Equation (44) is minimized:(44)$$e=\left|-\tan{(}^{e}\theta_{\mathrm{qmax}})-\frac{x_{h}\sin{(}^{h}\theta_{p}^{\prime }{-}^{h}\theta_{pi})+z_{h}\cos{(}^{h}\theta_{p}^{\prime }{-}^{h}\theta_{pi})}{x_{h}\cos{(}^{h}\theta_{p}^{\prime }{-}^{h}\theta_{pi})-z_{h}\sin{(}^{h}\theta_{p}^{\prime }{-}^{h}\theta_{pi})-d_{x}}\right|$$

Using Equations (37) and (38), ^h^*θ*_pq_ and ^w^*θ*_pq_ can be obtained.

#### 3.2.3. Vertical Rotation Angle Calculation

When the target’s *y* coordinate in the head coordinate system is greater than *d_y_*, the current angle of the head in the vertical direction is ^h^*θ*_t*i*_, and when the head is rotated in the vertical direction to ${}^{h}\theta_{t}^{\prime }$, the target is in the new head coordinate system. The 3D coordinates are
(45)$$\left(\begin{array}{c} \begin{array}{c} x_{h}^{\prime }\\ y_{h}^{\prime }\\ z_{h}^{\prime } \end{array}\\ 1 \end{array}\right)=\left(\begin{array}{cccc} 1 & 0 & 0 & 0\\ 0 & \cos{(}^{h}\theta_{t}^{\prime }{-}^{h}\theta_{ti}) & \sin{(}^{h}\theta_{t}^{\prime }{-}^{h}\theta_{ti}) & 0\\ 0 & -\sin{(}^{h}\theta_{t}^{\prime }{-}^{h}\theta_{ti}) & \cos{(}^{h}\theta_{t}^{\prime }{-}^{h}\theta_{ti}) & 0\\ 0 & 0 & 0 & 1 \end{array}\right)\left(\begin{array}{c} \begin{array}{c} x_{h}\\ y_{h}\\ z_{h} \end{array}\\ 1 \end{array}\right)$$

Therefore,
(46)$$\left(\begin{array}{c} x_{h}^{\prime }\\ y_{h}^{\prime }\\ z_{h}^{\prime } \end{array}\right)=\left(\begin{array}{c} x_{h}\\ y_{h}\cos{(}^{h}\theta_{t}^{\prime }{-}^{h}\theta_{ti})+z_{h}\sin{(}^{h}\theta_{t}^{\prime }{-}^{h}\theta_{ti})\\ z_{h}\cos{(}^{h}\theta_{t}^{\prime }{-}^{h}\theta_{ti})-y_{h}\sin{(}^{h}\theta_{t}^{\prime }{-}^{h}\theta_{ti}) \end{array}\right)$$

Using Equation (29), the coordinates of the eye coordinate system after the rotation of the target can be calculated:(47)$$\left(\begin{array}{c} {}^{e}x_{h}^{\prime }\\ {}^{e}y_{h}^{\prime }\\ {}^{e}z_{h}^{\prime } \end{array}\right)=\left(\begin{array}{c} d_{x}+x_{h}\\ y_{h}\cos{(}^{h}\theta_{t}^{\prime }{-}^{h}\theta_{ti})+z_{h}\sin{(}^{h}\theta_{t}^{\prime }{-}^{h}\theta_{ti})+d_{y}\\ z_{h}\cos{(}^{h}\theta_{t}^{\prime }{-}^{h}\theta_{ti})-y_{h}\sin{(}^{h}\theta_{t}^{\prime }{-}^{h}\theta_{ti}) \end{array}\right)$$

After rotation, the left and right motion modules reach the rotatable maximum value ^e^*θ*_tmax_ in the vertical direction, so that
(48)$$\tan{(}^{e}\theta_{\mathrm{tmax}})=\frac{{}^{e}y_{h}^{\prime }}{{}^{e}z_{h}^{\prime }}=\frac{y_{h}\cos{(}^{h}\theta_{t}^{\prime }{-}^{h}\theta_{ti})+z_{h}\sin{(}^{h}\theta_{t}^{\prime }{-}^{h}\theta_{ti})+d_{y}}{z_{h}\cos{(}^{h}\theta_{t}^{\prime }{-}^{h}\theta_{ti})-y_{h}\sin{(}^{h}\theta_{t}^{\prime }{-}^{h}\theta_{ti})}$$

Simplifying Equation (48), we obtain
(49)$$\sin{(}^{h}\theta_{t}^{\prime }{-}^{h}\theta_{ti})=\frac{z_{h}\tan{(}^{e}\theta_{\mathrm{tmax}})-y_{h}}{z_{h}+y_{h}\tan{(}^{e}\theta_{\mathrm{tmax}})}\cos{(}^{h}\theta_{t}^{\prime }{-}^{h}\theta_{ti})-\frac{d_{y}}{z_{h}+y_{h}\tan{(}^{e}\theta_{\mathrm{tmax}})}$$

Let
(50)$$\left\{\begin{array}{l} k_{1}=\frac{z_{h}\tan{(}^{e}\theta_{\mathrm{tmax}})-y_{h}}{z_{h}+y_{h}\tan{(}^{e}\theta_{\mathrm{tmax}})}\\ k_{2}=-\frac{d_{y}}{z_{h}+y_{h}\tan{(}^{e}\theta_{\mathrm{tmax}})} \end{array}\right.$$

Therefore,
(51)$${}^{h}\theta_{t}^{\prime }{=}^{h}\theta_{ti}+\arccos(\frac{-k_{1}k_{2}\pm \sqrt{k_{1}^{2}-k_{2}^{2}+1}}{k_{1}^{2}+1})$$

Equation (51) has two solutions; therefore, we choose the solution in which the deviation *e* of Equation (52) is minimized:(52)$$e=\left|\tan{(}^{e}\theta_{\mathrm{tmax}})-\frac{y_{h}\cos{(}^{h}\theta_{t}^{\prime }{-}^{h}\theta_{ti})+z_{h}\sin{(}^{h}\theta_{t}^{\prime }{-}^{h}\theta_{ti})+d_{y}}{z_{h}\cos{(}^{h}\theta_{t}^{\prime }{-}^{h}\theta_{ti})-y_{h}\sin{(}^{h}\theta_{t}^{\prime }{-}^{h}\theta_{ti})}\right|$$

Similarly, when the target’s *y* coordinates in the head coordinate system are less than *d_y_*, we have
(53)$$\tan({-}^{e}\theta_{\mathrm{tmax}})=\frac{{}^{e}y_{h}^{\prime }}{{}^{e}z_{h}^{\prime }}=\frac{y_{h}\cos{(}^{h}\theta_{t}^{\prime }{-}^{h}\theta_{ti})+z_{h}\sin{(}^{h}\theta_{t}^{\prime }{-}^{h}\theta_{ti})+d_{y}}{z_{h}\cos{(}^{h}\theta_{t}^{\prime }{-}^{h}\theta_{ti})-y_{h}\sin{(}^{h}\theta_{t}^{\prime }{-}^{h}\theta_{ti})}$$
(54)$$\sin{(}^{h}\theta_{t}^{\prime }{-}^{h}\theta_{ti})=-\frac{z_{h}\tan{(}^{e}\theta_{\mathrm{tmax}})+y_{h}}{z_{h}-y_{h}\tan{(}^{e}\theta_{\mathrm{tmax}})}\cos{(}^{h}\theta_{t}^{\prime }{-}^{h}\theta_{ti})-\frac{d_{y}}{z_{h}-y_{h}\tan{(}^{e}\theta_{\mathrm{tmax}})}$$
(55)$$\left\{\begin{array}{l} k_{1}=-\frac{z_{h}\tan{(}^{e}\theta_{\mathrm{tmax}})+y_{h}}{z_{h}-y_{h}\tan{(}^{e}\theta_{\mathrm{tmax}})}\\ k_{2}=-\frac{d_{y}}{z_{h}-y_{h}\tan{(}^{e}\theta_{\mathrm{tmax}})} \end{array}\right.$$
(56)$$e=\left|-\tan{(}^{e}\theta_{\mathrm{tmax}})-\frac{y_{h}\cos{(}^{h}\theta_{t}^{\prime }{-}^{h}\theta_{ti})+z_{h}\sin{(}^{h}\theta_{t}^{\prime }{-}^{h}\theta_{ti})+d_{y}}{z_{h}\cos{(}^{h}\theta_{t}^{\prime }{-}^{h}\theta_{ti})-y_{h}\sin{(}^{h}\theta_{t}^{\prime }{-}^{h}\theta_{ti})}\right|$$

When the obtained ${}^{h}\theta_{t}^{\prime }$ is outside of the range [−^h^*θ*_tmax_, ^h^*θ*_tmax_], the value of ^h^*θ*_tq_ is
(57)$${}^{h}\theta_{\mathrm{tq}}=\left\{\begin{array}{lll} {}^{h}\theta_{\mathrm{tmax}} & , & {}^{h}\theta_{t}^{\prime }\geq^{h}\theta_{\mathrm{tmax}}\\ {-}^{h}\theta_{\mathrm{tmax}} & , & {}^{h}\theta_{t}^{\prime }\leq {-}^{h}\theta_{\mathrm{tmax}}\\ {}^{h}\theta_{t}^{\prime } & , & \mathrm{else} \end{array}\right.$$

After obtaining ^h^*θ*_pq_, ^h^*θ*_tq_ and ^w^*θ*_pq_, $P_{e}^{\prime }\left(x_{e}^{\prime },y_{e}^{\prime },z_{e}^{\prime }\right)$ are the coordinates of the target in the eye coordinate system after the mobile robot and the head are rotated:(58)$$\left(\begin{array}{c} \begin{array}{c} x_{e}^{\prime }\\ y_{e}^{\prime }\\ z_{e}^{\prime } \end{array}\\ 1 \end{array}\right)=\left(\begin{array}{cc} \mathrm{Rot}(X{,}^{h}\theta_{\mathrm{tq}})\mathrm{Rot}(Y{,}^{h}\theta_{\mathrm{pq}}) & 0\\ 0 & 1 \end{array}\right)\left(\begin{array}{cc} \mathrm{Rot}(Y{,}^{w}\theta_{\mathrm{pq}}) & 0\\ 0 & 1 \end{array}\right)\left(\begin{array}{c} \begin{array}{c} x_{w}\\ y_{w}\\ z_{w} \end{array}\\ 1 \end{array}\right)+\left(\begin{array}{c} d_{x}\\ d_{y}\\ 0\\ 0 \end{array}\right)$$

The desired observation pose of the eye, characterized by ^l^*θ*_tq_, ^l^*θ*_pq_, ^r^*θ*_tq_ and ^r^*θ*_pq_, can be obtained using the method described in the following section. 

#### 3.2.4. Calculation of the Desired Observation Poses of the Eye

According to Formula (2), ^l^*θ*_tq_ = ^r^*θ*_tq_ = *θ*_t_, and ^l^*θ*_pq_ = ^r^*θ*_pq_ = *θ*_p_.

The inverse of the hand–eye matrix of the left camera and left motion module end coordinate system is
(59)$${}^{l}T_{m}^{-1}=\left(\begin{array}{cccc} {}^{l}n_{x} & {}^{l}o_{x} & {}^{l}a_{x} & {}^{l}p_{x}\\ {}^{l}n_{y} & {}^{l}o_{y} & {}^{l}a_{y} & {}^{l}p_{y}\\ {}^{l}n_{z} & {}^{l}o_{z} & {}^{l}a_{z} & {}^{l}p_{z}\\ 0 & 0 & 0 & 1 \end{array}\right)$$

The coordinate ^l^***P***_c_ (^l^*x*_c_, ^l^*y*_c_, ^l^*z*_c_) of $P_{e}^{\prime }\left(x_{e}^{\prime },y_{e}^{\prime },z_{e}^{\prime }\right)$ in the left camera coordinate system satisfies the following relationship:(60)$$\left(\begin{array}{c} {}^{l}P_{c}\\ 1 \end{array}\right)={}^{l}T_{m}^{-1}\left(\begin{array}{cc} \mathrm{Rot}(X,-{}^{l}\theta_{t}) & 0\\ 0 & 1 \end{array}\right)\left(\begin{array}{cc} \mathrm{Rot}(Y,-{}^{l}\theta_{p}) & 0\\ 0 & 1 \end{array}\right)\left(\begin{array}{c} P_{e}^{\prime }\\ 1 \end{array}\right)$$

According to the small hole imaging model, the imaging coordinates of the $P_{e}^{\prime }\left(x_{e}^{\prime },y_{e}^{\prime },z_{e}^{\prime }\right)$ point in the left camera are
(61)$$\left(\begin{array}{c} {}^{l}u\\ {}^{l}v\\ 1 \end{array}\right)={}^{l}M_{\mathrm{in}}{}^{l}P_{1c}=\left(\begin{array}{ccc} {}^{l}k_{x} & 0 & {}^{l}u{}_{0}\\ 0 & {}^{l}k_{y} & {}^{l}v{}_{0}\\ 0 & 0 & 1 \end{array}\right)\left(\begin{array}{c} {}^{l}x_{c}/{}^{l}z_{c}\\ {}^{l}y_{c}/{}^{l}z_{c}\\ 1 \end{array}\right)=\left(\begin{array}{c} {}^{l}k_{x}{}^{l}x_{c}/{}^{l}z_{c}+{}^{l}u{}_{0}\\ {}^{l}k_{y}{}^{l}y_{c}/{}^{l}z_{c}+{}^{l}v{}_{0}\\ 1 \end{array}\right)$$

Substituting Equation (61) into Equation (2), we obtain
(62)$$\left(\begin{array}{c} {}^{l}\Delta u\\ {}^{l}\Delta v \end{array}\right)=\left(\begin{array}{c} {}^{l}k_{x}{}^{l}x_{c}/{}^{l}z_{c}\\ {}^{l}k_{y}{}^{l}y_{c}/{}^{l}z_{c}\\ 1 \end{array}\right)$$

Based on the same principle, the coordinate ^r^***P***_c_ (^r^*x*_c_, ^r^*y*_c_, ^r^*z*_c_) of $P_{e}^{\prime }\left(x_{e}^{\prime },y_{e}^{\prime },z_{e}^{\prime }\right)$ in the right camera coordinate system is
(63)$$\left(\begin{array}{c} {}^{r}P_{c}\\ 1 \end{array}\right)={}^{r}T_{m}^{-1}\left(\begin{array}{cc} \mathrm{Rot}(X,-{}^{r}\theta_{t}) & 0\\ 0 & 1 \end{array}\right)\left(\begin{array}{cc} \mathrm{Rot}(Y,-{}^{r}\theta_{p}) & 0\\ 0 & 1 \end{array}\right){}^{r}T{}_{e}\left(\begin{array}{c} P_{e}^{\prime }\\ 1 \end{array}\right)$$

The imaging coordinates of point $P_{e}^{\prime }\left(x_{e}^{\prime },y_{e}^{\prime },z_{e}^{\prime }\right)$ in the right camera are
(64)$$\left(\begin{array}{c} {}^{r}u\\ {}^{r}v\\ 1 \end{array}\right)={}^{r}M_{\mathrm{in}}{}^{r}P_{1c}=\left(\begin{array}{ccc} {}^{r}k_{x} & 0 & {}^{r}u{}_{0}\\ 0 & {}^{r}k_{y} & {}^{r}v{}_{0}\\ 0 & 0 & 1 \end{array}\right)\left(\begin{array}{c} {}^{r}x_{c}/{}^{r}z_{c}\\ {}^{r}y_{c}/{}^{r}z_{c}\\ 1 \end{array}\right)=\left(\begin{array}{c} {}^{r}k_{x}{}^{r}x_{c}/{}^{r}z_{c}+{}^{r}u{}_{0}\\ {}^{r}k_{y}{}^{r}y_{c}/{}^{r}z_{c}+{}^{r}v{}_{0}\\ 1 \end{array}\right)$$
(65)$$\left(\begin{array}{c} {}^{r}\Delta u\\ {}^{r}\Delta v \end{array}\right)=\left(\begin{array}{c} {}^{r}k_{x}{}^{r}x_{c}/{}^{r}z_{c}\\ {}^{r}k_{y}{}^{r}y_{c}/{}^{r}z_{c}\\ 1 \end{array}\right)$$

By Equations (2), (62) and (65), two equations related to *θ*_t_ and *θ*_p_ (see Appendix C for the complete equations) can be obtained. It is challenging to calculate the values of *θ*_t_ and *θ*_p_ directly from these two equations, however. To obtain a solution, we consider a suboptimal observation pose and use this pose as the initial value; then, we use the trial-and-error method to obtain the optimal observation pose. When *θ*_t_ is calculated, let *θ*_p_ = 0; the solution of *θ*_t_ can then be obtained by Δ*v*_l_ = −Δ*v*_r_. When *θ*_p_ is calculated, the solution of *θ*_p_ is solved by Δ*u*_l_ = −Δ*u*_r_. The solution ***P***_ob_ (*θ*_t_, *θ*_t_, *θ*_p_, *θ*_p_) is a suboptimal observed pose. Based on the suboptimal observation pose, the trial-and-error method can be used to obtain the optimal solution with the smallest error. The range of *θ*_t_ is [−*θ*_tmax_, *θ*_tmax_]. The range of *θ*_p_ is [−*θ*_pmax_, *θ*_pmax_].

According to Equations (60) and (63), let *θ*_p_ be equal to 0 to obtain
(66)$$\left(\begin{array}{c} {}^{l}P_{c}\\ 1 \end{array}\right)={\left({}^{l}T_{m}\right)}^{-1}\left(\begin{array}{cc} \mathrm{Rot}(X,-\theta_{t}) & 0\\ 0 & 1 \end{array}\right){}^{l}T_{e}\left(\begin{array}{c} P_{e}^{\prime }\\ 1 \end{array}\right)$$

The following result is also available:(67)$$\left(\begin{array}{c} {}^{r}P_{c}\\ 1 \end{array}\right)={\left({}^{r}T_{m}\right)}^{-1}\left(\begin{array}{cc} \mathrm{Rot}(X,-\theta_{t}) & 0\\ 0 & 1 \end{array}\right){}^{r}T_{e}\left(\begin{array}{c} P_{e}^{\prime }\\ 1 \end{array}\right)$$

The base coordinate system of the left motion module is the world coordinate system. Therefore, ^l^***T***_w_ is a unit matrix. To simplify the calculation, we have
(68)$$\left(\begin{array}{c} {}^{r}P_{e}\\ 1 \end{array}\right)=\left(\begin{array}{c} {}^{r}x_{e}^{\prime }\\ {}^{r}y_{e}^{\prime }\\ {}^{r}z_{e}^{\prime }\\ 1 \end{array}\right)={}^{r}T_{e}\left(\begin{array}{c} P_{e}^{\prime }\\ 1 \end{array}\right)$$

According to the calculation principle of Section 3.2.1, we have the following:
(69)$$\left\{\begin{array}{l} \Delta u_{l}={}^{l}f_{x}\frac{\left({}^{l}o_{x}y_{e}^{\prime }+{}^{l}a_{x}z_{e}^{\prime })\cos\theta_{t}+({}^{l}o_{x}z_{e}^{\prime }-{}^{l}a_{x}y_{e}^{\prime })\sin\theta_{t}+({}^{l}p_{x}+{}^{l}n_{x}x_{e}^{\prime }\right)}{\left({}^{l}o_{z}y_{e}^{\prime }+{}^{l}a_{z}z_{e}^{\prime })\cos\theta_{t}+({}^{l}o_{z}z_{e}^{\prime }-{}^{l}a_{z}y_{e}^{\prime })\sin\theta_{t}+({}^{l}p_{z}+{}^{l}n_{z}x_{e}^{\prime }\right)}\\ \Delta v_{l}={}^{l}f_{y}\frac{\left({}^{l}o_{y}y_{e}^{\prime }+{}^{l}a_{y}z_{e}^{\prime })\cos\theta_{t}+({}^{l}o_{y}z_{e}^{\prime }-{}^{l}a_{y}y_{e}^{\prime })\sin\theta_{t}+({}^{l}p_{y}+{}^{l}n_{y}x_{e}^{\prime }\right)}{\left({}^{l}o_{z}y_{e}^{\prime }+{}^{l}a_{z}z_{e}^{\prime })\cos\theta_{t}+({}^{l}o_{z}z_{e}^{\prime }-{}^{l}a_{z}y_{e}^{\prime })\sin\theta_{t}+({}^{l}p_{z}+{}^{l}n_{z}x_{e}^{\prime }\right)} \end{array}\right.$$
(70)$$\left\{\begin{array}{l} \Delta u_{r}={}^{r}f_{x}\frac{\left({}^{r}o_{x}{}^{r}y_{e}^{\prime }+{}^{r}a_{x}{}^{r}z_{e}^{\prime })\cos\theta_{t}+({}^{r}o_{x}{}^{r}z_{e}^{\prime }-{}^{r}a_{x}{}^{r}y_{e}^{\prime })\sin\theta_{t}+({}^{r}p_{x}+{}^{r}n_{x}{}^{r}x_{e}^{\prime }\right)}{\left({}^{r}o_{z}{}^{r}y_{e}^{\prime }+{}^{r}a_{z}{}^{r}z_{e}^{\prime })\cos\theta_{t}+({}^{r}o_{z}z_{e}^{\prime }-{}^{r}a_{z}{}^{r}y_{e}^{\prime })\sin\theta_{t}+({}^{r}p_{z}+{}^{r}n_{z}{}^{r}x_{e}^{\prime }\right)}\\ \Delta v_{r}={}^{r}f_{y}\frac{\left({}^{r}o_{y}{}^{r}y_{e}^{\prime }+{}^{r}a_{y}{}^{r}z_{e}^{\prime })\cos\theta_{t}+({}^{r}o_{y}{}^{r}z_{e}^{\prime }-{}^{r}a_{y}{}^{r}y_{e}^{\prime })\sin\theta_{t}+({}^{r}p_{y}+{}^{r}n_{y}{}^{r}x_{e}^{\prime }\right)}{\left({}^{r}o_{z}{}^{r}y_{e}^{\prime }+{}^{r}a_{z}{}^{r}z_{e}^{\prime })\cos\theta_{t}+({}^{r}o_{z}{}^{r}z_{e}^{\prime }-{}^{r}a_{z}{}^{r}y_{e}^{\prime })\sin\theta_{t}+({}^{r}p_{z}+{}^{r}n_{z}{}^{r}x_{e}^{\prime }\right)} \end{array}\right.$$


Assume the following:(71)$$E_{\mathrm{sv}}=\left|\Delta v_{l}\right|+\left|\Delta v_{r}\right|$$

The solution to *θ*_t_ that keeps the target at the center of the two cameras needs to satisfy the following conditions:(72)$$\left\{\begin{array}{l} \Delta v_{l}+\Delta v_{r}=0\\ -\theta_{\mathrm{tmax}}\leq \theta_{t}\leq \theta_{\mathrm{tmax}}\\ \theta_{t}=\arg\min(E_{\mathrm{sv}}) \end{array}\right.$$

Substituting the second equation of Equations (69) and (70) into Equation (72) and solving the equation, we have
(73)$$k_{1}\cos^{2}\theta_{t}+k_{2}\sin^{2}\theta_{t}+k_{3}\sin\theta_{t}\cos\theta_{t}+k_{4}\cos\theta_{t}+k_{5}\sin\theta_{t}+k_{6}=0$$
where $k_{1},k_{2},k_{3},k_{4},k_{5}$ are
(74)$$k_{1}={}^{l}f_{y}({}^{l}o_{y}y_{e}^{\prime }+{}^{l}a_{y}z_{e}^{\prime })({}^{r}o_{z}{}^{r}y_{e}^{\prime }+{}^{r}a_{z}{}^{r}z_{e}^{\prime })+{}^{r}f_{y}({}^{l}o_{z}y_{e}^{\prime }+{}^{l}a_{z}z_{e}^{\prime })({}^{r}o_{y}{}^{r}y_{e}^{\prime }+{}^{r}a_{y}{}^{r}z_{e}^{\prime })$$
(75)$$k_{2}={}^{l}f_{y}({}^{l}o_{y}z_{e}^{\prime }-{}^{l}a_{y}y_{e}^{\prime })({}^{r}o_{z}{}^{r}z_{e}^{\prime }-{}^{r}a_{z}{}^{r}y_{e}^{\prime })+{}^{r}f_{y}({}^{l}o_{z}z_{e}^{\prime }-{}^{l}a_{z}y_{e}^{\prime })({}^{r}o_{y}{}^{r}z_{e}^{\prime }-{}^{r}a_{y}{}^{r}y_{e}^{\prime })$$
(76)$$\begin{array}{r} k_{3}={}^{l}f_{y}({}^{l}o_{y}y_{e}^{\prime }+{}^{l}a_{y}z_{e}^{\prime })({}^{r}o_{z}{}^{r}z_{e}^{\prime }-{}^{r}a_{z}{}^{r}y_{e}^{\prime })+{}^{l}f_{y}({}^{l}o_{y}z_{e}^{\prime }-{}^{l}a_{y}y_{e}^{\prime })({}^{r}o_{z}{}^{r}y_{e}^{\prime }+{}^{r}a_{z}{}^{r}z_{e}^{\prime })\\ +{}^{r}f_{y}({}^{l}o_{z}y_{e}^{\prime }+{}^{l}a_{z}z_{e}^{\prime })({}^{r}o_{y}{}^{r}z_{e}^{\prime }-{}^{r}a_{y}{}^{r}y_{e}^{\prime })+{}^{r}f_{y}({}^{l}o_{z}z_{e}^{\prime }-{}^{l}a_{z}y_{e}^{\prime })({}^{r}o_{y}{}^{r}y_{e}^{\prime }+{}^{r}a_{y}{}^{r}z_{e}^{\prime }) \end{array}$$
(77)$$\begin{array}{r} k_{4}={}^{l}f_{y}({}^{l}o_{y}y_{e}^{\prime }+{}^{l}a_{y}z_{e}^{\prime })({}^{r}p_{z}+{}^{r}n_{z}{}^{r}x_{e}^{\prime })+{}^{l}f_{y}({}^{l}p_{y}+{}^{l}n_{y}x_{e}^{\prime })({}^{r}o_{z}{}^{r}y_{e}^{\prime }+{}^{r}a_{z}{}^{r}z_{e}^{\prime })\\ +{}^{r}f_{y}({}^{l}o_{z}y_{e}^{\prime }+{}^{l}a_{z}z_{e}^{\prime })({}^{r}p_{y}+{}^{r}n_{y}{}^{r}x_{e}^{\prime })+{}^{r}f_{y}({}^{l}p_{z}+{}^{l}n_{z}x_{e}^{\prime })({}^{r}o_{y}{}^{r}y_{e}^{\prime }+{}^{r}a_{y}{}^{r}z_{e}^{\prime }) \end{array}$$
(78)$$\begin{array}{r} k_{5}={}^{l}f_{y}({}^{l}o_{y}z_{e}^{\prime }-{}^{l}a_{y}y_{e}^{\prime })({}^{r}p_{z}+{}^{r}n_{z}{}^{r}x_{e}^{\prime })+{}^{l}f_{y}({}^{l}p_{y}+{}^{l}n_{y}x_{e}^{\prime })({}^{r}o_{z}{}^{r}z_{e}^{\prime }-{}^{r}a_{z}{}^{r}y_{e}^{\prime })\\ +{}^{r}f_{y}({}^{l}o_{z}z_{e}^{\prime }-{}^{l}a_{z}y_{e}^{\prime })({}^{r}p_{y}+{}^{r}n_{y}{}^{r}x_{e}^{\prime })+{}^{r}f_{y}({}^{l}p_{z}+{}^{l}n_{z}x_{e}^{\prime })({}^{r}o_{y}{}^{r}z_{e}^{\prime }-{}^{r}a_{y}{}^{r}y_{e}^{\prime }) \end{array}$$
(79)$$k_{6}={}^{l}f_{y}({}^{l}p_{y}+{}^{l}n_{y}x_{e}^{\prime })({}^{r}p_{z}+{}^{r}n_{z}{}^{r}x_{e}^{\prime })+{}^{r}f_{y}({}^{l}p_{z}+{}^{l}n_{z}x_{e}^{\prime })({}^{r}p_{y}+{}^{r}n_{y}{}^{r}x_{e}^{\prime })$$

According to the triangle relationship, we have
(80)$$\cos^{2}\theta_{t}+\sin^{2}\theta_{t}=1$$

Replacing cos*θ*_t_ in Equation (73) with sin*θ*_t_, we obtain the following:(81)$$k_{1}^{\prime }\sin^{4}\theta_{t}+k_{2}^{\prime }\sin^{3}\theta_{t}+k_{3}^{\prime }\sin^{2}\theta_{t}+k_{4}^{\prime }\sin\theta_{t}+k_{5}^{\prime }=0$$
where $k_{1},k_{2},k_{3},k_{4},k_{5}$ are
(82)$$k_{1}^{\prime }={\left(k_{2}-k_{1}\right)}^{2}+k_{3}^{2}$$
(83)$$k_{2}^{\prime }=2(k_{2}-k_{1})k_{5}+2k_{3}k_{4}$$
(84)$$k_{3}^{\prime }=2(k_{2}-k_{1})k_{6}+k_{5}^{2}+k_{4}^{2}-k_{3}^{2}$$
(85)$$k_{4}^{\prime }=2k_{5}k_{6}-2k_{3}k_{4}$$
(86)$$k_{5}^{\prime }=k_{6}^{2}-k_{4}^{2}$$

Four solutions can be obtained using Equation (81). The optimal solution is a real number, and the most suitable solution can be selected by the condition of Equation (72).

After *θ*_t_ is obtained, *θ*_p_ can be solved based on the obtained *θ*_t_.

According to Equations (60) and (63), ${\bar{\theta }}_{t}$ is the solution obtained in Section 3.2.2, so that
(87)$$\left(\begin{array}{c} {}^{l}P_{c}\\ 1 \end{array}\right)={\left({}^{l}T_{m}\right)}^{-1}\left(\begin{array}{cc} \mathrm{Rot}(X,-{\bar{\theta }}_{t}) & 0\\ 0 & 1 \end{array}\right)\left(\begin{array}{cc} \mathrm{Rot}(Y,-{}^{l}\theta_{p}) & 0\\ 0 & 1 \end{array}\right){}^{l}T_{e}\left(\begin{array}{c} P_{e}^{\prime }\\ 1 \end{array}\right)$$

The following result is also available:(88)$$\left(\begin{array}{c} {}^{r}P_{c}\\ 1 \end{array}\right)={\left({}^{r}T_{m}\right)}^{-1}\left(\begin{array}{cc} \mathrm{Rot}(X,-{\bar{\theta }}_{t}) & 0\\ 0 & 1 \end{array}\right)\left(\begin{array}{cc} \mathrm{Rot}(Y,-{}^{l}\theta_{p}) & 0\\ 0 & 1 \end{array}\right){}^{r}T_{e}\left(\begin{array}{c} P_{e}^{\prime }\\ 1 \end{array}\right)$$

Since ${\bar{\theta }}_{t}$ is known, for convenience of calculation, we set
(89)$${}^{l}T_{m}^{\prime }={\left({}^{l}T_{m}\right)}^{-1}\left(\begin{array}{cc} \mathrm{Rot}(X,-{\bar{\theta }}_{t}) & 0\\ 0 & 1 \end{array}\right)=\left(\begin{array}{cccc} {}^{l}n_{x}^{\prime } & {}^{l}o_{x}^{\prime } & {}^{l}a_{x}^{\prime } & {}^{l}p_{x}^{\prime }\\ {}^{l}n_{y}^{\prime } & {}^{l}o_{y}^{\prime } & {}^{l}a_{y}^{\prime } & {}^{l}p_{y}^{\prime }\\ {}^{l}n_{z}^{\prime } & {}^{l}o_{z}^{\prime } & {}^{l}a_{z}^{\prime } & {}^{l}p_{z}^{\prime }\\ 0 & 0 & 0 & 1 \end{array}\right)$$
(90)$${}^{r}T_{m}^{\prime }={\left({}^{r}T_{m}\right)}^{-1}\left(\begin{array}{cc} \mathrm{Rot}(X,-{\bar{\theta }}_{t}) & 0\\ 0 & 1 \end{array}\right)=\left(\begin{array}{cccc} {}^{r}n_{x}^{\prime } & {}^{r}o_{x}^{\prime } & {}^{r}a_{x}^{\prime } & {}^{r}p_{x}^{\prime }\\ {}^{r}n_{y}^{\prime } & {}^{r}o_{y}^{\prime } & {}^{r}a_{y}^{\prime } & {}^{r}p_{y}^{\prime }\\ {}^{r}n_{z}^{\prime } & {}^{r}o_{z}^{\prime } & {}^{r}a_{z}^{\prime } & {}^{r}p_{z}^{\prime }\\ 0 & 0 & 0 & 1 \end{array}\right)$$

The following results are obtained:
(91)$$\left\{\begin{array}{l} \Delta u_{l}={}^{l}f_{x}\frac{\left({}^{l}n_{x}^{\prime }x_{e}^{\prime }+{}^{l}a_{x}^{\prime }z_{e}^{\prime })\cos\theta_{p}+({}^{l}a_{x}^{\prime }x_{e}^{\prime }-{}^{l}n_{x}^{\prime }z_{e}^{\prime })\sin\theta_{p}+({}^{l}p_{x}^{\prime }+{}^{l}o_{x}^{\prime }y_{e}^{\prime }\right)}{\left({}^{l}n_{z}^{\prime }x_{e}^{\prime }+{}^{l}a_{z}^{\prime }z_{e}^{\prime })\cos\theta_{p}+({}^{l}a_{z}^{\prime }x_{e}^{\prime }-{}^{l}n_{z}^{\prime }z_{e}^{\prime })\sin\theta_{p}+({}^{l}p_{z}^{\prime }+{}^{l}o_{z}^{\prime }y_{e}^{\prime }\right)}\\ \Delta v_{l}={}^{l}f_{y}\frac{\left({}^{l}n_{y}^{\prime }x_{e}^{\prime }+{}^{l}a_{y}^{\prime }z_{e}^{\prime })\cos\theta_{p}+({}^{l}a_{y}^{\prime }x_{e}^{\prime }-{}^{l}n_{y}^{\prime }z_{e}^{\prime })\sin\theta_{p}+({}^{l}p_{y}^{\prime }+{}^{l}o_{y}^{\prime }y_{e}^{\prime }\right)}{\left({}^{l}n_{z}^{\prime }x_{e}^{\prime }+{}^{l}a_{z}^{\prime }z_{e}^{\prime })\cos\theta_{p}+({}^{l}a_{z}^{\prime }x_{e}^{\prime }-{}^{l}n_{z}^{\prime }z_{e}^{\prime })\sin\theta_{p}+({}^{l}p_{z}^{\prime }+{}^{l}o_{z}^{\prime }y_{e}^{\prime }\right)} \end{array}\right.$$
(92)$$\left\{\begin{array}{l} \Delta u_{r}={}^{r}f_{x}\frac{\left({}^{r}n_{x}^{\prime }{}^{r}x_{e}^{\prime }+{}^{r}a_{x}^{\prime }{}^{r}z_{e}^{\prime })\cos\theta_{p}+({}^{r}a_{x}^{\prime }{}^{r}x_{e}^{\prime }-{}^{r}n_{x}^{\prime }{}^{r}z_{e}^{\prime })\sin\theta_{p}+({}^{r}p_{x}^{\prime }+{}^{r}o_{x}^{\prime }{}^{r}y_{e}^{\prime }\right)}{\left({}^{r}n_{z}^{\prime }{}^{r}x_{e}^{\prime }+{}^{r}a_{z}^{\prime }{}^{r}z_{e}^{\prime })\cos\theta_{p}+({}^{r}a_{z}^{\prime }{}^{r}x_{e}^{\prime }-{}^{r}n_{z}^{\prime }{}^{r}z_{e}^{\prime })\sin\theta_{p}+({}^{r}p_{z}^{\prime }+{}^{r}o_{z}^{\prime }{}^{r}y_{e}^{\prime }\right)}\\ \Delta v_{r}={}^{r}f_{y}\frac{\left({}^{r}n_{y}^{\prime }{}^{r}x_{e}^{\prime }+{}^{r}a_{y}^{\prime }{}^{r}z_{e}^{\prime })\cos\theta_{p}+({}^{r}a_{y}^{\prime }{}^{r}x_{e}^{\prime }-{}^{r}n_{y}^{\prime }{}^{r}z_{e}^{\prime })\sin\theta_{p}+({}^{r}p_{y}^{\prime }+{}^{r}o_{y}^{\prime }{}^{r}y_{e}^{\prime }\right)}{\left({}^{r}n_{z}^{\prime }{}^{r}x_{e}^{\prime }+{}^{r}a_{z}^{\prime }{}^{r}z_{e}^{\prime })\cos\theta_{p}+({}^{r}a_{z}^{\prime }{}^{r}x_{e}^{\prime }-{}^{r}n_{z}^{\prime }{}^{r}z_{e}^{\prime })\sin\theta_{p}+({}^{r}p_{z}^{\prime }+{}^{r}o_{z}^{\prime }{}^{r}y_{e}^{\prime }\right)} \end{array}\right.$$

Assume that
(93)$$E_{\mathrm{su}}=\left|\Delta u_{l}\right|+\left|\Delta u_{r}\right|$$

The solution to *θ*_p_ that keeps the target at the center of the two cameras needs to satisfy the following conditions:(94)$$\left\{\begin{array}{l} \Delta u_{l}+\Delta u_{r}=0\\ -\theta_{\mathrm{pmax}}\leq \theta_{p}\leq \theta_{\mathrm{pmax}}\\ \theta_{p}=\arg\min(E_{\mathrm{su}}) \end{array}\right.$$

Substituting the second equation of Equations (91) and (92) into Equation (94) and solving the available equation, we obtain
(95)$$k_{1}\cos^{2}\theta_{p}+k_{2}\sin^{2}\theta_{p}+k_{3}\sin\theta_{p}\cos\theta_{p}+k_{4}\cos\theta_{p}+k_{5}\sin\theta_{p}+k_{6}=0$$
where
(96)$$k_{1}={}^{l}f_{x}({}^{l}n_{x}^{\prime }x_{e}^{\prime }+{}^{l}a_{x}^{\prime }z_{e}^{\prime })({}^{r}n_{z}^{\prime }{}^{r}x_{e}^{\prime }+{}^{r}a_{z}^{\prime }{}^{r}z_{e}^{\prime })+{}^{r}f_{x}({}^{r}n_{x}^{\prime }{}^{r}x_{e}^{\prime }+{}^{r}a_{x}^{\prime }{}^{r}z_{e}^{\prime })({}^{l}n_{z}^{\prime }x_{e}^{\prime }+{}^{l}a_{z}^{\prime }z_{e}^{\prime })$$
(97)$$k_{2}={}^{l}f_{x}({}^{l}n_{x}^{\prime }z_{e}^{\prime }-{}^{l}a_{x}^{\prime }x_{e}^{\prime })({}^{r}n_{z}^{\prime }{}^{r}z_{e}^{\prime }-{}^{r}a_{z}^{\prime }{}^{r}x_{e}^{\prime })+{}^{r}f_{x}({}^{r}n_{x}^{\prime }{}^{r}z_{e}^{\prime }-{}^{r}a_{x}^{\prime }{}^{r}x_{e}^{\prime })({}^{l}n_{z}^{\prime }z_{e}^{\prime }-{}^{l}a_{z}^{\prime }x_{e}^{\prime })$$
(98)$$\begin{array}{r} k_{3}={}^{l}f_{x}({}^{l}n_{x}^{\prime }x_{e}^{\prime }+{}^{l}a_{x}^{\prime }z_{e}^{\prime })({}^{r}n_{z}^{\prime }{}^{r}z_{e}^{\prime }-{}^{r}a_{z}^{\prime }{}^{r}x_{e}^{\prime })+{}^{l}f_{x}({}^{l}n_{x}^{\prime }z_{e}^{\prime }-{}^{l}a_{x}^{\prime }x_{e}^{\prime })({}^{r}n_{z}^{\prime }{}^{r}x_{e}^{\prime }+{}^{r}a_{z}^{\prime }{}^{r}z_{e}^{\prime })\\ +{}^{r}f_{x}({}^{r}n_{x}^{\prime }{}^{r}z_{e}^{\prime }-{}^{r}a_{x}^{\prime }{}^{r}x_{e}^{\prime })({}^{l}n_{z}^{\prime }x_{e}^{\prime }+{}^{l}a_{z}^{\prime }z_{e}^{\prime })+{}^{r}f_{x}({}^{r}n_{x}^{\prime }{}^{r}x_{e}^{\prime }+{}^{r}a_{x}^{\prime }{}^{r}z_{e}^{\prime })({}^{l}n_{z}^{\prime }z_{e}^{\prime }-{}^{l}a_{z}^{\prime }x_{e}^{\prime }) \end{array}$$
(99)$$\begin{array}{r} k_{4}={}^{l}f_{x}({}^{l}n_{x}^{\prime }x_{e}^{\prime }+{}^{l}a_{x}^{\prime }z_{e}^{\prime })({}^{r}o_{z}^{\prime }{}^{r}y_{e}^{\prime }+{}^{r}p_{z}^{\prime })+{}^{l}f_{x}({}^{l}o_{x}^{\prime }y_{e}^{\prime }+{}^{l}p_{x}^{\prime })({}^{r}n_{z}^{\prime }{}^{r}x_{e}^{\prime }+{}^{r}a_{z}^{\prime }{}^{r}z_{e}^{\prime })\\ +{}^{r}f_{x}({}^{r}o_{x}^{\prime }{}^{r}y_{e}^{\prime }+{}^{r}p_{x}^{\prime })({}^{l}n_{z}^{\prime }x_{e}^{\prime }+{}^{l}a_{z}^{\prime }z_{e}^{\prime })+{}^{r}f_{x}({}^{r}n_{x}^{\prime }{}^{r}x_{e}^{\prime }+{}^{r}a_{x}^{\prime }{}^{r}z_{e}^{\prime })({}^{l}o_{z}^{\prime }y_{e}^{\prime }+{}^{l}p_{z}^{\prime }) \end{array}$$
(100)$$\begin{array}{r} k_{5}={}^{l}f_{x}({}^{l}n_{x}^{\prime }z_{e}^{\prime }-{}^{l}a_{x}^{\prime }x_{e}^{\prime })({}^{r}o_{z}^{\prime }{}^{r}y_{e}^{\prime }+{}^{r}p_{z}^{\prime })+{}^{l}f_{x}({}^{l}o_{x}^{\prime }y_{e}^{\prime }+{}^{l}p_{x}^{\prime })({}^{r}n_{z}^{\prime }{}^{r}z_{e}^{\prime }-{}^{r}a_{z}^{\prime }{}^{r}x_{e}^{\prime })\\ +{}^{r}f_{x}({}^{r}o_{x}^{\prime }{}^{r}y_{e}^{\prime }+{}^{r}p_{x}^{\prime })({}^{l}n_{z}^{\prime }z_{e}^{\prime }-{}^{l}a_{z}^{\prime }x_{e}^{\prime })+{}^{r}f_{x}({}^{r}n_{x}^{\prime }{}^{r}z_{e}^{\prime }-{}^{r}a_{x}^{\prime }{}^{r}x_{e}^{\prime })({}^{l}o_{z}^{\prime }y_{e}^{\prime }+{}^{l}p_{z}^{\prime }) \end{array}$$
(101)$$k_{6}={}^{l}f_{x}({}^{l}o_{x}^{\prime }y_{e}^{\prime }+{}^{l}p_{x}^{\prime })({}^{r}o_{z}^{\prime }{}^{r}y_{e}^{\prime }+{}^{r}p_{z}^{\prime })+{}^{r}f_{x}({}^{r}o_{x}^{\prime }{}^{r}y_{e}^{\prime }+{}^{r}p_{x}^{\prime })({}^{l}o_{z}^{\prime }y_{e}^{\prime }+{}^{l}p_{z}^{\prime })$$

Replacing cos*θ*_p_ in Equation (73) with sin*θ*_p_, we obtain
(102)$$k_{1}^{\prime }\sin^{4}\theta_{p}+k_{2}^{\prime }\sin^{3}\theta_{p}+k_{3}^{\prime }\sin^{2}\theta_{p}+k_{4}^{\prime }\sin\theta_{p}+k_{5}^{\prime }=0$$
where
(103)$$k_{1}^{\prime }={\left(k_{2}-k_{1}\right)}^{2}+{\left(k_{3}\right)}^{2}$$
(104)$$k_{2}^{\prime }=2(k_{2}-k_{1})k_{5}+2k_{3}k_{4}$$
(105)$$k_{3}^{\prime }=2(k_{2}-k_{1})k_{6}+{\left(k_{5}\right)}^{2}+{\left(k_{4}\right)}^{2}-{\left(k_{3}\right)}^{2}$$
(106)$$k_{4}^{\prime }=2k_{5}k_{6}-2k_{3}k_{4}$$
(107)$$k_{5}^{\prime }={\left(k_{6}\right)}^{2}-{\left(k_{4}\right)}^{2}$$

Four solutions can be obtained using Equation (102). The optimal solution must be a real number, and the most suitable solution can be selected using the condition of Equation (94). For the case where the four solutions cannot satisfy Equation (94), the position of the target is beyond the position that the bionic eye can reach. In this case, compensation is required through the head or torso. ${\bar{\theta }}_{t}$ and ${\bar{\theta }}_{p}$ obtained at this time are suboptimal solutions close to the optimal solution. *θ*_t_ and *θ*_p_ are the optimal solutions.

Through the above steps, the desired observation pose can be calculated. The calculation steps of ^g^***f***_q_ can be summarized by the flow chart shown in Figure 8.

### 3.3. Desired Pose Calculation for Approaching Gaze Point Tracking

The mobile robot approaches the target in two steps: the first step is that the robot and the head rotate in the horizontal direction until the robot and the head are facing the target, and the second step is that the robot moves straight toward the target. The desired position of the approaching motion should satisfy the following conditions: (1) the target should be on the *Z* axis of the robot and the head coordinate system, (2) the distance between the target and the robot should be less than the set threshold *D*_T_ and (3) the eye should be in the optimal observation position. ^a^***f***_q_ can be defined as
(108)$${}^{a}f_{q}=\left(\begin{array}{l} {}^{w}x_{\mathrm{mq}}=0\\ {}^{w}z_{\mathrm{mq}}=\{z\left|\right.\;\;0<{}^{w}z_{m}-z\leq D_{T}\}\\ {}^{w}\theta_{\mathrm{pq}}=\{\theta \left|\right.\;\;|\theta |\leq 2\pi {,}^{w}\gamma_{p}=0\}\\ {}^{h}\theta_{\mathrm{pq}}=0\\ {}^{h}\theta_{\mathrm{tq}}=\{\theta \left|\right.\;\;|\theta |\leq {}^{h}\theta_{\mathrm{tmax}},{|}^{h}\gamma_{t}|\leq^{h}\gamma_{\mathrm{tmax}}\}\\ {}^{l}\theta_{\mathrm{pq}}={}^{r}\theta_{\mathrm{pq}}=\{\theta \left|\right.\;\;|\theta |\leq {}^{e}\theta_{\mathrm{pmax}},\Delta m_{l}=-\Delta m_{r}\}\\ {}^{l}\theta_{\mathrm{tq}}={}^{r}\theta_{\mathrm{tq}}=\{\theta \left|\right.\;\;|\theta |\leq {}^{e}\theta_{\mathrm{tmax}},\Delta m_{l}=-\Delta m_{r}\} \end{array}\right)$$

The desired rotation angle ^w^*θ*_pq_ of the moving robot is the same as the angle *^b^γ*_p_ between the robot and the target and can be obtained by
(109)$${}^{w}\theta_{\mathrm{pq}}={}^{w}\gamma_{p}=\arctan(\frac{{}^{w}z_{m}}{{}^{w}x_{m}})$$
^h^*θ*_tq_ can be obtained using the method described in Section 3.2. The optimal observation pose described in Section 3.2.4 can be used to obtain ^l^*θ*_tq_, ^l^*θ*_pq_, ^r^*θ*_tq_ and ^r^*θ*_pq_.

## 4. Robot Pose Control

After obtaining the desired pose of the robot system, the control block diagram shown in Figure 9 is used to control the robot to move to the desired pose.

The desired pose is converted to the desired position of the motor. Δ*θ*_lt_, Δ*θ*_lp_, Δ*θ*_rt_, Δ*θ*_rp_, Δ*θ*_ht_ and Δ*θ*_hp_ are deviations of the desired angle from the current angle of motor M_lu_, motor M_ld_, motor M_ru_, motor M_rd_, motor M_hu_ and motor M_hd_, respectively. ^l^*θ*_m_ and ^r^*θ*_m_ are the angles at which each wheel of the moving robot needs to be rotated. During the in situ gaze point tracking process, the moving robot performs only the rotation of the original position, and the angle of the robot movement can be calculated according to the desired angle of the robot. When the robot rotates, the two wheels move in opposite directions at the same speed. Let the distance between the two wheels of the moving robot be *D*_r_; when the robot rotates around an angle ^w^*θ*_pq_, the distance that each wheel needs to move is
(110)$$S={}^{w}\theta_{\mathrm{pq}}\frac{D_{r}}{2}$$

The diameter of each wheel is *d*_w_, and the angle of rotation of each wheel is (where counterclockwise is positive)
(111)$${}^{r}\theta_{m}=-{}^{l}\theta_{m}=\frac{2S}{d_{w}}$$

In the process of approaching the target, the moving robot follows a straight line, and the angle of rotation of each wheel is
(112)$${}^{r}\theta_{m}={}^{l}\theta_{m}=\frac{2{}^{w}z_{\mathrm{mq}}}{d_{w}}$$

The movement of the moving robot is achieved by controlling the rotation of each wheel. Each wheel is equipped with a DC brushless motor, and a DSP2000 controller is used to control the movement of the DC brushless motor. Position servo control is implemented in the DSP2000 controller. 

In the robot system, the weight of the camera and lens is approximately 80 g, the weight of the camera and the fixed mechanical parts is approximately 50 g and the motor that controls the vertical rotation of the camera (rotating around the horizontal axis of rotation) and the corresponding encoder weighs approximately 250 g. The mechanical parts of the fixed vertical rotating motor and encoder weigh approximately 100 g. The radius of the rotation of the camera in the vertical direction is approximately 1 cm, and the rotation in the horizontal direction (rotation about the vertical axis of rotation) has a radius of approximately 2 cm. Therefore, when the gravitational acceleration is 9.8 m/s^2^, the torque required for the vertical rotating electric machine is approximately 0.013 N·m, and the torque required for the horizontal rotating electric machine is approximately 0.043 N·m. The vertical rotating motor uses a 28BYG5401 stepping motor with a holding torque of 0.1 N·m and a positioning torque of 0.008 N·m. The driver is HSM20403A. The horizontal rotating motor is a 57BYGH301 stepping motor with a holding torque of 1.5 N·m, a positioning torque of 0.07 N·m and drive model HSM20504A. The four stepping motors of the eye have a step angle of 1.8° and are all subdivided by 25, so the actual step angle of each motor is 0.072°, and the minimum pulse width that the driver can receive is 2.5 µs. The stepper motor has a maximum angular velocity of 200°/s.

The head vertical rotary motor uses a 57BYGH401 stepper motor with a holding torque of 2.2 N·m, a positioning torque of 0.098 N·m and drive model HSM20504A. The head horizontal rotary motor is an 86BYG350B three-phase AC stepping motor with a holding torque of 5 N·m, a positioning torque of 0.3 N-m and an HSM30860M driver. The step angle of the head motor after subdivision is also 0.072°. The head vertical motor has a load of approximately 5 kg and a radius of rotation of less than 1 cm. The head horizontal rotary motor has a load of approximately 9.5 kg and a radius of rotation of approximately 5 cm. In the experiment, we found that the maximum horizontal pulse frequency that the head horizontal rotary motor can receive is 0.6 Kpps. Its maximum angular velocity is 43.2°/s.

## 5. Experiments and Discussion

Using the robot platform introduced in Section 2, experiments on in situ gaze point tracking and approaching gaze point tracking were performed

Each camera has a resolution of 400 × 300 pixels. The directions of rotation are [−45°, 45°]. The range of rotation of the head is [−30°, 30°]. *d_x_* and *d_y_* are 150 mm and 200 mm, respectively. The internal and external parameters, distortion parameters, initial position parameters and left- and right-hand–eye parameters of the dual purpose method are calibrated as follows:(113)$${}^{l}M_{\mathrm{in}}=\left(\begin{array}{ccc} 341.58 & 0 & 201.6\\ 0 & 341.97 & 147.62\\ 0 & 0 & 1 \end{array}\right)$$
(114)$$K_{l}=\left(\begin{array}{ccccc} -0.1905 & 0.2171 & -0.0018 & -0.0005 & -0.0823 \end{array}\right)$$
(115)$${}^{l}T_{m}=\left(\begin{array}{cccc} 1.0 & 0.0078 & -0.0022 & 58.4172\\ 0.0001 & 0.9954 & 0.0959 & 3.6042\\ \;0.0013 & -0.0959 & 0.9954 & 51.9366\\ 0 & 0 & 0 & 1 \end{array}\right)$$
(116)$${}^{r}M_{\mathrm{in}}=\left(\begin{array}{ccc} 335.13 & 0 & 184.32\\ 0 & 335.5 & 141.26\\ 0 & 0 & 1 \end{array}\right)$$
(117)$$K_{r}=\left(\begin{array}{ccccc} -0.1861 & 0.1987 & -0.004 & -0.0011 & -0.0739 \end{array}\right)$$
(118)$${}^{r}T_{m}=\left(\begin{array}{cccc} 0.9999 & -0.0086 & -0.0125 & -45.0147\\ 0.0190 & -0.9969 & -0.0782 & -24.5528\\ \;0.0097 & -0.0784 & 0.9970 & 42.9270\\ 0 & 0 & 0 & 1 \end{array}\right)$$
(119)$${}^{l}T_{r}=\left(\begin{array}{cccc} 0.9998 & -0.0099 & -0.0193 & 189.5922\\ 0.0095 & 0.9997 & -0.0215 & -0.0426\\ \;0.0195 & 0.0213 & 0.9996 & 8.9671\\ 0 & 0 & 0 & 1 \end{array}\right)$$

The experimental in situ gaze point tracking scene is shown in Figure 10, with a checkerboard target used as the target. For in situ gaze point tracking, the target is held by a person. In the approaching target gaze tracking experiment, the target is fixed in front of the robot.

### 5.1. In Situ Gaze Point Tracking Experiment

In the in situ gaze experiment, the target moves at a low speed within a certain range, and the robot combines the movement of the eye, the head and the mobile robot so that the binocular vision can always perceive the 3D coordinates of the target at the optimal observation posture. This experiment prompts the robot to find the target and gaze at it. In the gaze point tracking process, binocular stereo vision is used to calculate the 3D coordinates of the target in the eye coordinate system in real time. Through the positional relationship between the eye and the head, the coordinate system of the target in the eye can be converted to the head coordinate system. Similarly, the 3D coordinates of the target in the robot coordinate system can be obtained. Through the 3D coordinates, the desired poses of the eyes, head and mobile robot are calculated according to the method proposed in this paper. Then, the camera is controlled to the desired position by the stepping motor; after reaching the desired position, the image and the motor position information are collected again, and the 3D coordinates of the target are calculated. 

In the experiment, the angles between the head and the target, ^h^*γ*_pmax_ and ^h^*γ*_pmax_, are each 30°. The method described in Section 3 is used to calculate the desired pose of each joint of the robot based on the 3D coordinates of the target. In the experiment, the actual coordinate position and desired coordinate position of the target in the binocular image space, the actual position and desired position of the eye and head motor, the angle between the head and the robot and the target, and the target in the robot coordinate system are stored. Figure 11a,b show the *u* and *v* coordinates of the target on the left image, respectively, and Figure 11c,d show the *u* and *v* coordinates of the target on the right image, respectively. The desired image coordinates are recalculated based on the optimal observation position. Figure 11e–h show the positions of the tilt motor (M_lu_) of the left eye, the pan motor (M_ld_) of the left eye, the tilt motor (M_ru_) of the right eye and the pan motor (M_rd_) of the right eye, respectively. Figure 11i shows the positions of the pan motor (M_hd_) of the head. Since the target moves in the vertical direction with small amplitude, the motor M_hu_ does not rotate, and the case is similar to the motion principle of the motor M_hd_, so the motor position of the head only provides the result of the motor M_hd_. Figure 11j shows the angle deviation and rotation. In this figure, T-h is the angle between the head and target, T-r is the angle between the robot and target, R-r is the angle of the robot rotation from the origin location and T-o is the angle of the target to the origin location. Figure 11k shows the coordinates (*^w^x*, *^w^z*) of the target in the world coordinate system. Figure 11l shows the coordinates (*^o^x*, *^o^z*) of the target in the world coordinate system of the origin location.

As shown in Figure 11, the image coordinate of the target is substantially within ±40 pixels in the central region of the left and right images in the *x* direction. These coordinates are kept within ±10 pixels of the center region of the left and right images in the *y* direction. Throughout the experiment, the target was rotated approximately 200° around the robot. The robot moved approximately 140°, the head rotated 30° and the target could be kept in the center region of the binocular images. The motor position curve shows that the motor’s operating position can track the desired position very well. The angle variation curve shows that the angle between the target and the head and the robot changes and that the robot turning angles are suitably consistent. The coordinates of the target shown in Figure 11 in the robot coordinate system and the coordinates of the target in the initial position of the world coordinate system are very close to the actual position change in the target’s position.

Through the above analysis, we can determine the following: (1) It is feasible to realize gaze point tracking of a robot based on 3D coordinates. (2) Using the movement of the head, eyes and mobile robot used in this paper, it is possible to achieve gaze point tracking of the target while ensuring minimum resource consumption.

### 5.2. Approaching Gaze Point Tracking Experiment

The approaching gaze point tracking experimental scene is shown in Figure 12.

The robot approaches the target without obstacles and reaches the area in which the robot can operate on the target. The target can be grasped or carefully observed. In the approaching gaze experiment, a target is fixed at a position 2.2 m from the robot, and when the robot moves to a position where the distance from the target to robot is 0.6 m, the motion is stopped, and the maximum speed of the moving robot is 1 m/s. The experiment realizes the approaching movement to the target in two steps: first, the head, the eye and the moving robot chassis are rotated so that the head and the moving robot are facing the target, and the head observes the target in the optimal observation posture; second, the movement is controlled. The robot moves linearly in the target’s direction. During the movement, the angles of the head and the eye are fine-tuned, and the 3D coordinates of the target are detected in real time until the *z* coordinate of the target in the robot coordinate system is less than the threshold set to stop the motion.

Figure 13 shows the results of the approaching gaze point tracking experiment. Figure 13a,b show the *u* and *v* coordinates of the target on the left image, respectively, and Figure 13c,d show the *u* and *v* coordinates of the target on the right image, respectively. The desired image coordinates are recalculated based on the optimal observation position. Figure 13e–h show the positions of the tilt motor (M_lu_) of the left eye, the pan motor (M_ld_) of the left eye, the tilt motor (M_ru_) of the right eye and the pan motor (M_rd_) of the right eye, respectively. Figure 13i shows the positions of the pan motor (M_hd_) of the head. Figure 13j shows the angle deviation and rotation. T-h is the angle between the head and the target, T-r is the angle between the robot and the target, R-r is the angle of the robot’s rotation from the origin location and T-o is the angle of the target to the origin location. Figure 13k shows the coordinates (*^w^x*, *^w^z*) of the target in the world coordinate system. Figure 13l shows the robot’s forward distance and the distance between the target and the robot. 

The change in the image’s coordinate curve indicates that the coordinates of the target in the left and right images move from the initial position to the central region of the image and stabilize in the center region of the image during the approach process. In the process of turning towards the target in the first step, the target coordinates in the image fluctuate because the head motor rotates a large amount and is accompanied by a certain vibration during the rotation, which can be avoided by using a system with better stability. The variety curve of the motor position in Figure 13 shows that the motion of the motor can track the target well with the desired pose, and the prediction of the 3D coordinates is not used during the tracking process, so this prediction is accompanied by a cycle lag. The changes in angle in Figure 13 show that the robot system achieves the task of steering toward the target in the first few control cycles and then moves toward the target at a stable angle. Figure 13a shows the change in the coordinates of the target in the robot coordinate system. When the robot rotates, fluctuations arise around the measured x coordinate, mainly due to the measurement error caused by the shaking of the system. The experimental results in Figure 13b show that the robot’s movement toward the target is very consistent. During the approach process, the target can be kept within ±50 pixels of the desired position in the horizontal direction of the image while being within ±20 pixels of the desired position in the vertical direction of the image. The eye motor achieves fast tracking of the target in 1.5 s. The angle between the target and the head is reduced from 20° to 0°, and the angle between the target and the robot is reduced from 35° to 0°. The robot then over-turns. At 34°, the target changes by 34° from the initial position.

Through the above analysis, it can be found that by using the combination of the head, the eye and the trunk in the present method, the approach toward the target can be achieved while ensuring that the robot is gazing at the target.

## 6. Conclusions

This study achieved gaze point tracking based on the 3D coordinates of the target. First, a robot experiment platform was designed. Based on the bionic eye experiment platform, a head with two degrees of freedom was added, using the mobile robot as a carrier. 

Based on the characteristics of the robot platform, this paper proposed a method of gaze point tracking. To achieve in situ gaze point tracking, the combination of the eyes, head and trunk is designed based on the principles of minimum resource consumption and maximum system stability. Eye rotation consumes the least amount of resources and has minimal impact on the stability of the overall system during the exercise. The head rotation consumes more resources than the eyeball but fewer than the trunk rotation. At the same time, the rotation of the head affects the stability of the eyeball but only minimally affects the stability of the entire robotic system. The resources consumed by the rotation of the trunk generally predominate, and the rotation of the trunk tends to affect the stability of the head and the eye. Therefore, when the eye can observe the target in the optimal observation posture, only the eye is rotated; otherwise, the head is rotated, and when the angle at which the head needs to move exceeds its threshold, the mobile robot rotates. When approaching gaze point tracking is performed, the robot and head first face the target and then move straight toward the vicinity of the target. Based on the proposed gaze point tracking method, this paper provides an expected pose calculation method for the horizontal rotation angle and the vertical rotation angle.

Based on the experimental robot platform, a series of experiments was performed, and the effectiveness of the gaze point tracking method was verified. In our future works, a practical task of delivering medicine in a hospital and more detailed comparative experiments, as well as discussions with other similar studies, will be implemented.

## Figures and Tables

**Figure 1 sensors-23-06299-f001:**
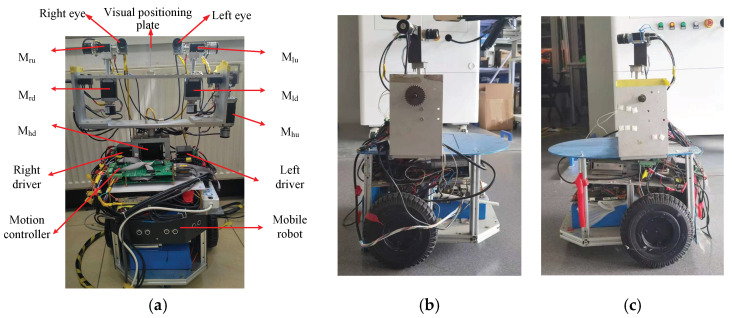
Physical implementation of the robot system. (**a**) The front side. (**b**) The left side. (**c**) The right side.

**Figure 2 sensors-23-06299-f002:**
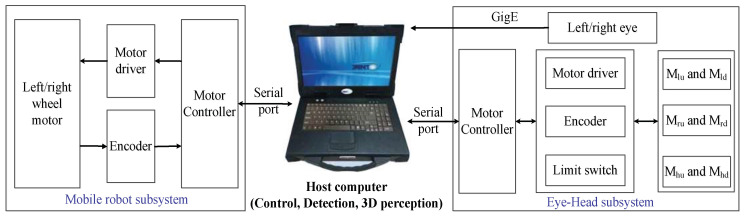
Robot system’s organization diagram.

**Figure 3 sensors-23-06299-f003:**
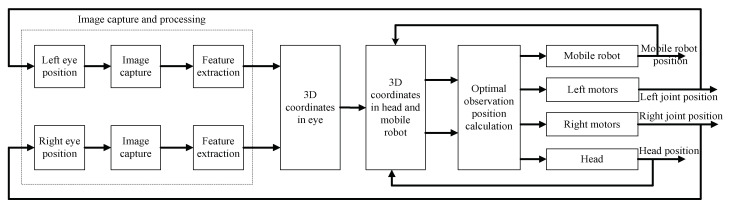
Block diagram of the gaze point tracking control system.

**Figure 4 sensors-23-06299-f004:**
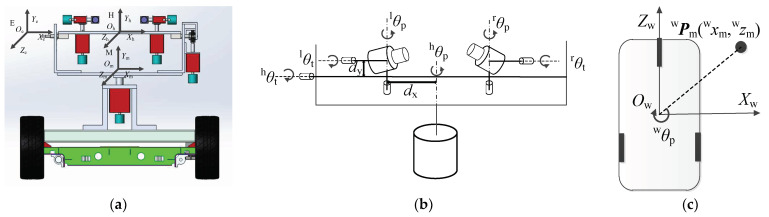
Robot coordinates system and system parameter definition, (**a**) coordinate system definition, (**b**) eye–head system parameters and (**c**) mobile robot parameters.

**Figure 5 sensors-23-06299-f005:**
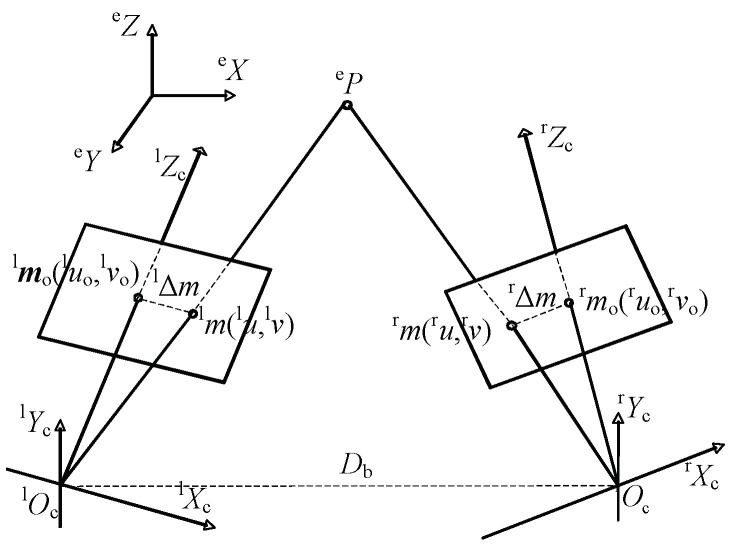
Schematic of the relationship between a Cartesian point and its image point.

**Figure 6 sensors-23-06299-f006:**
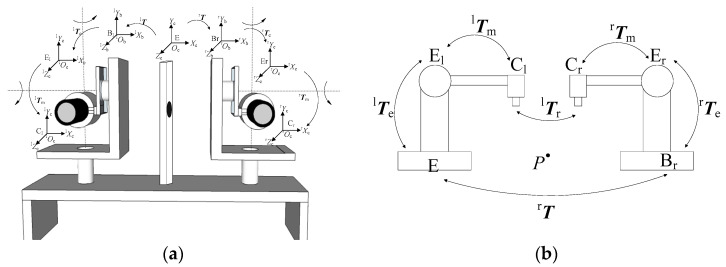
(**a**) Mechanical structure and coordinate systems of the bionic eye platform and (**b**) binocular 3D perception principle of bionic eyes.

**Figure 7 sensors-23-06299-f007:**
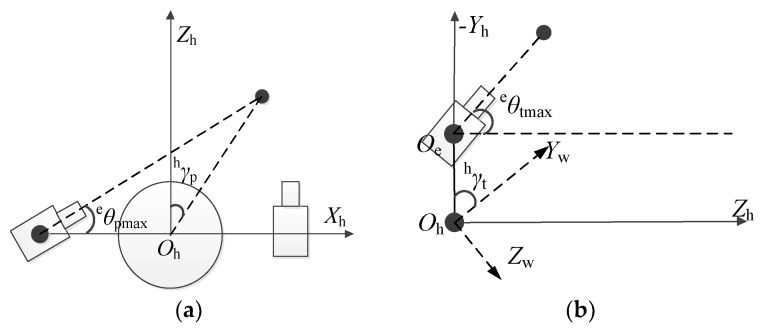
Principle of head rotation calculation in fixation point tracking: (**a**) horizontal rotation angle and (**b**) vertical rotation angle.

**Figure 8 sensors-23-06299-f008:**
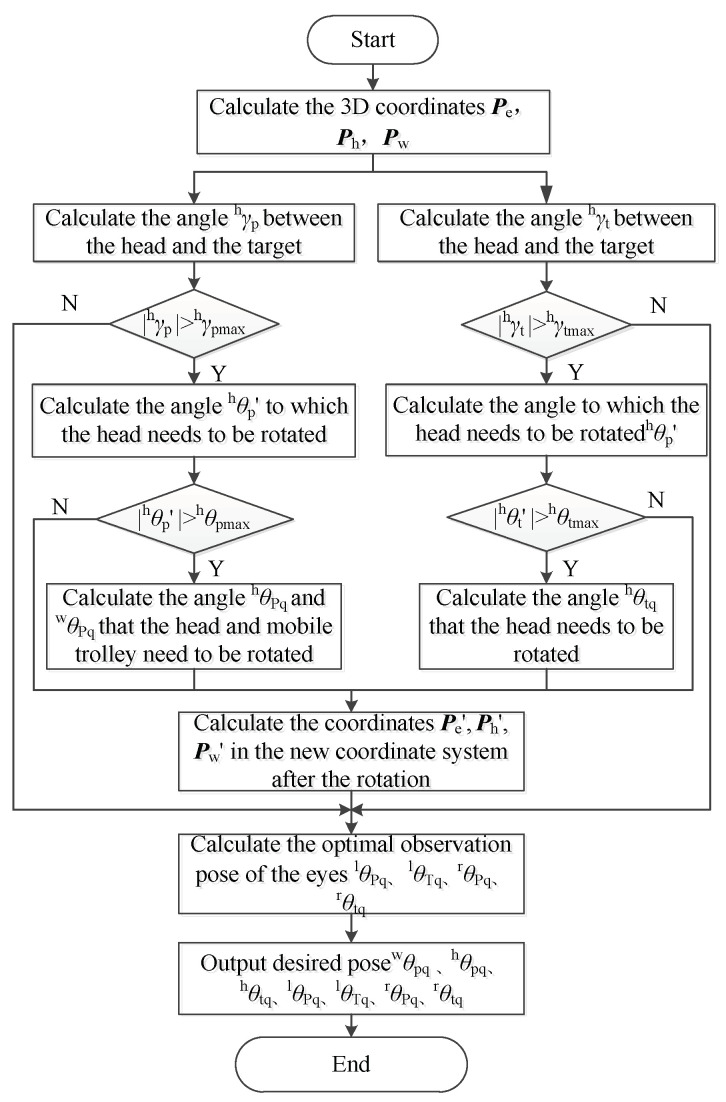
Steps for calculating the desired pose of the fixation point.

**Figure 9 sensors-23-06299-f009:**
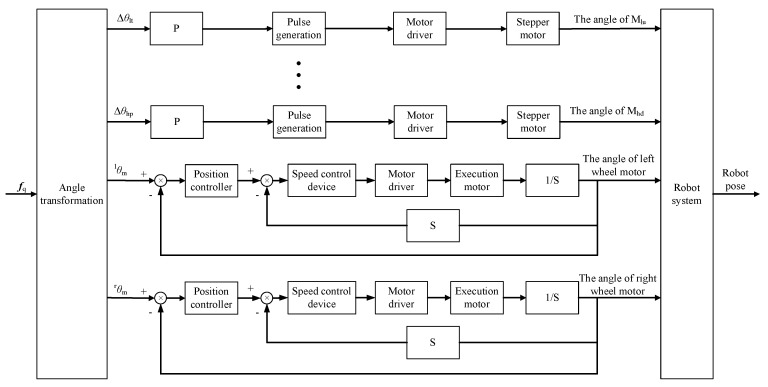
Robot pose control block diagram.

**Figure 10 sensors-23-06299-f010:**
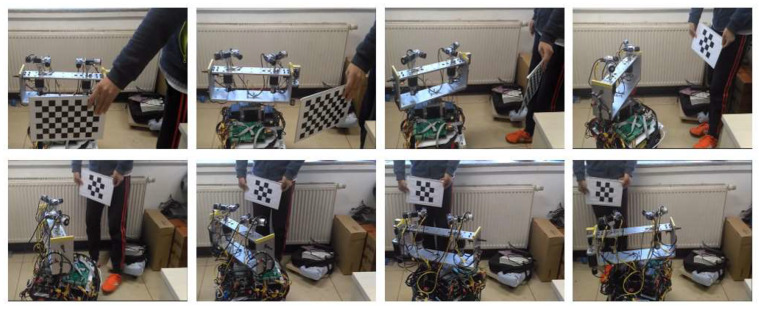
Experimental in situ gaze point tracking scene.

**Figure 11 sensors-23-06299-f011:**
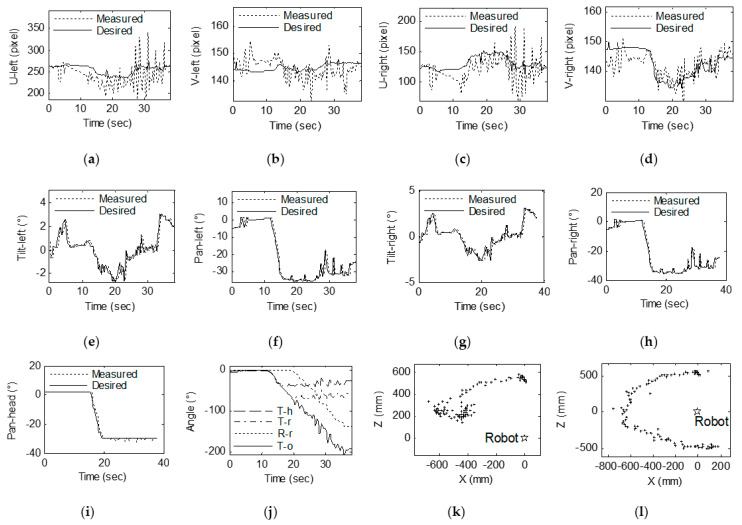
Experimental results of gaze shifting to the target: (**a**) *U* coordinates of the target on the left image. (**b**) *V* coordinates of the target on the left image. (**c**) *U* coordinates of the target on the right image. (**d**) *V* coordinates of the target on the right image. (**e**) Left camera tilt. (**f**) Left camera pan. (**g**) Right camera tilt. (**h**) Right camera pan. (**i**) Head pan. (**j**) Angle deviation and rotation. (**k**) Coordinates (*^w^x*, *^w^z*) of the target in the world coordinate system. (**l**) Coordinates (*^o^x*, *^o^z*) of the target in the world coordinate system based on the origin location. The “+” in the subfigures (**k**,**l**) represents the position of the target in the coordinate system, and the “☆” represents the position of the robot in the coordinate system.

**Figure 12 sensors-23-06299-f012:**
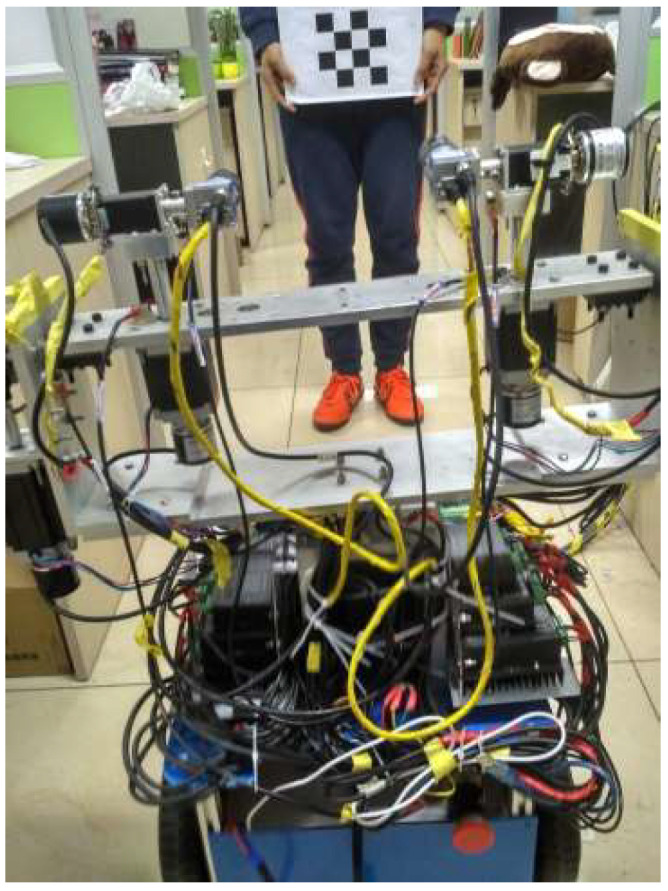
Experimental approaching gaze point tracking scene.

**Figure 13 sensors-23-06299-f013:**
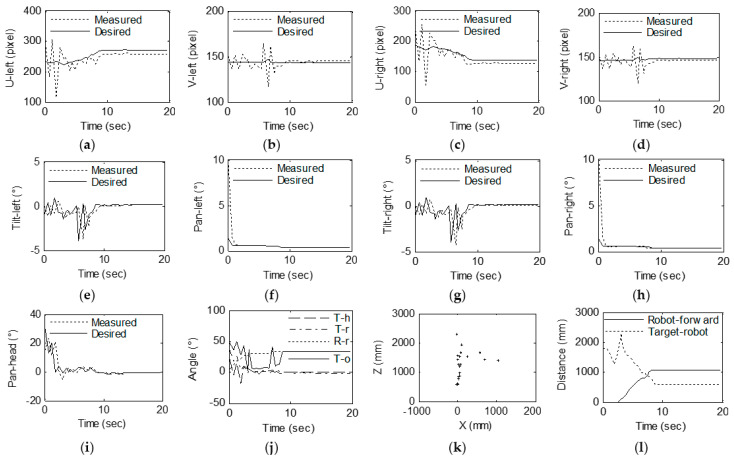
Experimental results of gaze shifting to the target: (**a**) *U* coordinates of the target on the left image. (**b**) *V* coordinates of the target on the left image. (**c**) *U* coordinates of the target on the right image. (**d**) *V* coordinates of the target on the right image. (**e**) Left camera tilt. (**f**) Left camera pan. (**g**) Right camera tilt. (**h**) Right camera pan. (**i**) Head pan. (**j**) Angular deviation and rotation. (**k**) Coordinates (*^w^x*, *^w^z*) of the target in the world coordinate system. (**l**) Robot forward distance and the distance between the target and robot.

## Data Availability

No new data was created.

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
