# Peer review of "Gaze Point Tracking Based on a Robotic Body–Head–Eye Coordination Method"

_sensors, 2023, doi:10.3390/s23146299_

Round 1

Reviewer 1 Report

Introduction

- In the second paragraph, the authors start the sentence like this “In [5] …” the authors must choose to mention the authors, or mention the study, for example “In a study [5]”, or if they mention the section of the study they intend to cite and at the end they cite the references. But as it stands, it is incorrect.

- The authors do this again in the third paragraph, with reference 9. Please correct to Kuang et al [9], in the fourth paragraph for references 12 and 14, and in the sixth paragraph for reference 16, please correct.

- The entire fifth paragraph is without any references, please reference the sentences.

- The introduction is well written and presents a logical and temporal sequence of ideas, however, despite explaining well the differences between previously developed robots and their “flaws or limitations” that the authors intend not to commit and control in this new project, something that I missed in the text was in relation to the practical application of the robot. The authors do not explain what their purpose is, what they are building the robot for. In the future, will these devices be able to help in the assessment of hand-eye coordination? Or help in the rehabilitation of patients with vestibular dysfunction, for example, who have nystagmus? I believe that the inclusion of this practical purpose of the robot would improve the reader's interest in the article and also in the device, and I suggest that the authors add something in this regard.

Platform and control system

- I suggest that the authors include two side photos of the robot, right and left, because, during the article, he mentions that the robot has two wheels and in figure 1 it is not possible to see the wheels. I suggest the authors call Figure 1 a (front photo), b and c (right and left side photos) of the robot.

- The authors mention the step-by-step of how the robot and its equipment were created, however, they do not mention any reference of where and why they used these parameters. I suggest authors reference these parameters.

Control system

- The same happens in this section, the authors mention the step-by-step of how the robot is controlled and its parameters, however, they do not mention any reference of where and why they used these parameters. I suggest authors reference these parameters.

Experiments and Discussion

- I missed the discussion of discussing what the authors found with similar studies in the literature.

- The authors only demonstrated the experiments of their study and demonstrated in which figures their results were. However, there is no discussion of these experiments nor in the sense of other similar studies, nor of how these experiments can be applied in humans.

- The discussion needs to be better directed towards the practical and clinical part of the robot. For what activities or area can these robots be implemented? The discussion should be improved in order to show what similar studies found with the results of this study and bring a practical utility to the robot and its experiments.

Reviewer 2 Report

This manuscript presents the design of robot system equipped with eyes and a head, and the in situ gaze point tracking and approaching gaze point tracking are studied in order to make a robot gaze at targets rapidly and stably.

After reviewing this manuscript, the following issues need to be improved:

1.     Besides the motor Mhd and Mhu(DOFs), there are two motors equipped with left and right eyes(2DOFs). As this is different from conventional configuration(only Mhd and Mhu, the pan has 2DOFs), please add the necessity of the design in detail.

2.     How to describe the rapidity and stability of the system, please add key parameters comparison with similar system.

3.     Please shorten the length of the manuscript.

Can be improved.

Round 2

Reviewer 1 Report

Congratulations to the authors, they did a good job, I believe that now the article is clearer for a better understanding of the readers.